

# 1 Validation of 10-year SAO OMI Ozone Profile (PROFOZ)
# 2 Product Using Ozonesonde Observations

Guanyu Huang[1,*], Xiong Liu[1], Kelly Chance[1], Kai Yang[2], Pawan K. Bhartia[3], Zhaonan Cai[1],
Marc Allaart[4], Bertrand Calpini[5], Gerrie J. R. Coetzee[6], Emilio Cuevas-Agulló[7], Manuel
Cupeiro[8] , Hugo De Backer[9], Manvendra K. Dubey[10], , Henry E. Fuelberg[11], Masatomo
Fujiwara[12], Sophie Godin-Beekmann[13], Tristan J. Hall[11], Bryan Johnson[14], Everette Joseph[15],
Rigel Kivi[16], Bogumil Kois[17], Ninong Komala[18], Gert König-Langlo[19], Giovanni Laneve[20],
Thierry Leblanc[21],  Marion Marchand[13], Kenneth R. Minschwaner[22], Gary Morris[23], Mike J.
Newchurch[24], Shin-Ya Ogino[25], Nozomu Ohkawara[26], Ankie J. M. Piters[4], Françoise Posny[27],
Richard Querel[28], Rinus Scheele[4], Frank J.  Schmidlin[3], Russell C. Schnell[14], Otto Schrems[19],
Henry Selkirk[29], Masato Shiotani[30], Pavla Skrivánková[31], René Stübi[5], Ghassan Taha[29],
David W. Tarasick[32], Anne M.  Thompson[3], Valérie Thouret[33], Matt Tully[34], Roeland van
Malderen[9], Geraint Vaughan[35], Holger Vömel[36], Peter von der Gathen[37], Jacquelyn C. Witte[38],
Margarita Yela[39]
1.  Harvard-Smithsonian Center for Astrophysics, Cambridge, MA, USA
2.  Department of Atmospheric and Oceanic Science, University of Maryland, College Park,
Maryland, USA
3.  NASA Goddard Space Flight Center, Greenbelt, Maryland, USA
4.  Royal Netherlands Meteorological Institute (KNMI), De Bilt, the Netherlands
5.  MeteoSwiss Aerological Station, Federal Office of Meteorology and Climatology
MeteoSwiss, Payerne, Switzerland
6.  South African Weather Service, Pretoria, South Africa
7.  Izana Atmospheric Research Center, Meteorological State Agency of Spain, Santa Cruz de
Tenerife, Spain
8.  National Meteorological Service, Ushuaia, Tierra del Fuego, Argentina
9.  Royal Meteorological Institute of Belgium, Brussel, Belgium
10. Los Alamos National Laboratory, Los Alamos, NM, USA
11. Earth, Ocean and Atmospheric Sciences, Florida State University, Tallahassee, FL, USA





12. Faculty of Environmental Earth Science, Hokkaido University, Sapporo, Japan
13. LATMOS-ISPL, Université Paris 6 Pierre-et-Marie-Curie, Paris, France
14. NOAA/ESRL Global Monitoring Division, Boulder, CO, USA
15. Atmospheric Sciences Research Center, SUNY University at Albany, Albany, NY, USA
16. Finnish Meteorological Institute, Helsinki, Finland
17. The Institute of Meteorology and Water Management, National Research Institute, Warsaw,
36       Poland

18. Indonesian Institute of Aeronautics and Space (LAPAN), Bandung, Indonesia
19. Alfred Wegener Institute for Polar and Marine Research, Bremerhaven, Germany
20. Earth Observation Satellite Images Applications Lab (EOSIAL), Università di Roma 'La
40       Sapienza', Rome, Italy

21. Jet Propulsion Laboratory, California Institute of Technology, Pasadena, CA, USA
22. Department of Physics, New Mexico Institute of Mining and Technology, Socorro, NM, USA
23. St. Edward's University, Austin, TX, USA
24. Department of Atmospheric Science, University of Alabama in Huntsville, Huntsville, AL,
45       USA

25. Department of Coupled Ocean-Atmosphere-Land Processes Research, Japan Agency for
47       Marine-Earth Science and Technology, Yokosuka, Japan

26. Global Environment and Marine Department, Japan Meteorological Agency, Tokyo, Japan
27. Université de la Réunion, Saint Denis, France
28. National Institute of Water and Atmospheric Research, Lauder, Central Otago, New Zealand
29. Universities Space Research Association, Greenbelt, MD, USA
30. Research Institute for Sustainable Humanosphere, Kyoto University, Kyoto, Japan
31. Upper Air and Surface Observation Department, Czech Hydrometeorological Institute, Praha,
54       Czech Republic

32. Air Quality Research Division, Environment & Climate Change Canada, Downsview, ON,
56       Canada.

33. Laboratoire d'Aerologie, Université de Toulouse, Toulouse, France
34. Observations & Infrastructure Division, Bureau of Meteorology, Melbourne, Victoria,
59       Australia



35. School of Earth, Atmosphere and Environmental Sciences, University of Manchester,

61       Manchester, U.K.

36. Earth Observing Laboratory, National Center for Atmospheric Research, Boulder, CO, USA
37. Alfred Wegener Institute, Potsdam, Germany
38. Science Systems and Applications Inc. Greenbelt, MD, USA
39. Atmospheric Research and Instrumentation Branch, National Institute for Aerospace

66       Technology (INTA), Madrid, Spain

*Correspondence to: Guanyu Huang (guanyu.huang@cfa.harvard.edu)





**Abstract**
We validate the Ozone Monitoring Instrument (OMI) ozone-profile (PROFOZ) product from
October 2004 through December 2014 retrieved by the Smithsonian Astrophysical Observatory
(SAO) algorithm against ozonesonde observations. We also evaluate the effects of OMI Row
anomaly (RA) on the retrieval by dividing the data set into before and after the occurrence of
serious OMI RA, i.e., pre-RA (2004-2008) and post-RA (2009-2014). The retrieval shows good
agreement with ozonesondes in the tropics and mid-latitudes and for pressure < ~50 hPa in the
high latitudes. It demonstrates clear improvement over the a priori down to the lower troposphere
in the tropics and down to an average of ~550 (300) hPa at middle (high latitudes). In the tropics
and mid-latitudes, the profile mean biases (MBs) are less than 6%, and the standard deviations
(SDs) range from 5-10% for pressure < ~50 hPa to less than 18% (27%) in the tropics (mid-
latitudes) for pressure > ~50 hPa after applying OMI averaging kernels to ozonesonde data. The
MBs of the stratospheric ozone column (SOC) are within 2% with SDs of < 5% and the MBs of
the tropospheric ozone column (TOC) are within 6% with SDs of 15%. In the high latitudes, the
profile MBs are within 10% with SDs of 5-15% for pressure < ~50 hPa, but increase to 30% with
SDs as great as 40% for pressure > ~50 hPa. The SOC MBs increase up to 3% with SDs as great
as 6% and the TOC SDs increase up to 30%. The comparison generally degrades at larger solar-
zenith angles (SZA) due to weaker signals and additional sources of error, leading to worse
performance at high latitudes and during the mid-latitude winter. Agreement also degrades with
increasing cloudiness for pressure > ~100 hPa and varies with cross-track position, especially with
large MBs and SDs at extreme off-nadir positions. In the tropics and mid-latitudes, the post-RA
comparison is considerably worse with larger SDs reaching 2% in the stratosphere and 8% in the
troposphere and up to 6% in TOC. There are systematic differences that vary with latitude
compared to the pre-RA comparison. The retrieval comparison demonstrates good long-term
stability during the pre-RA period, but exhibits a statistically significant trend of 0.14-0.7%/year
for pressure < ~ 80 hPa, 0.7 DU/year in SOC and -0.33 DU/year in TOC during the post-RA period.
The spatiotemporal variation of retrieval performance suggests the need to improve OMI's
radiometric calibration especially during the post-RA period to maintain the long-term stability
and reduce the latitude/season/SZA and cross-track dependence of retrieval quality.





## 1 Introduction

The Dutch-Finnish built Ozone Monitoring Instrument (OMI) on board the NASA Aura satellite
has been making useful measurements of trace gases including ozone and aerosols since October
2004. There are various retrieval algorithms to retrieve ozone profile and/or total ozone from OMI
data (Bak et al., 2015). The ozone profile retrieval algorithm initially developed at the Smithsonian
Astrophysical Observatory (Liu et al., 2005) for Global Ozone Monitoring Experiment (GOME)
data was adapted to OMI data (Liu et al., 2010b). Total ozone column (OC), Stratospheric Ozone
Column (SOC) and Tropospheric Ozone Column (TOC) can be directly derived from the retrieved
ozone profile with retrieval errors in in the range of a few Dobson Units (DU) (Liu et al., 2006b;
Liu et al., 2010a). This algorithm has been put into production in the OMI Science Investigator-
led Processing System (SIPS), processing the entire OMI data record with approximately one-
month delay. The ozone profile product titled PROFOZ is publicly available at the Aura Validation
Data Center (AVDC) (http://avdc.gsfc.nasa.gov/index.php?site=2045907950). This long-term
ozone profile product, with high spatial resolution and daily global coverage, constitutes a useful
dataset to study the spatial and temporal distribution of ozone.

To effectively use the retrieval dataset, it is necessary to evaluate and understand its retrieval
quality and long-term performance. Although validation of the ozone profile product (mostly
earlier versions) has been partially performed against aircraft, ozonesonde, and Microwave Limb
Sounder (MLS) data, these evaluations are limited to certain time periods and/or spatial region
and/or to only portion of the product (e.g., total ozone columns (OC) or TOC only) (Pittman et al.,
2009; Liu et al., 2010a; Liu et al., 2010b; Sellitto et al., 2011; Wang et al., 2011; Bak et al., 2013a;
Lal et al., 2013; Ziemke et al., 2014; Hayashida et al., 2015). Additionally, the quality of ozone
profile retrievals is very sensitive to the signal to noise ratio (SNR) of the radiance measurements
as well as their radiometric calibration, which may degrade over time as shown in GOME and
GOME-2 retrievals (Liu et al., 2007; Cai et al., 2012). Although OMI's optical degradation is
remarkably small to within 1-2% over the years, the SNR and the number of good spectral pixels
(not flagged as bad/hot pixels) have been gradually decreasing over the years due to the expected
CCD degradation (Claas, 2014). Furthermore, the occurrence of RA, which affects level 1b data
at all wavelengths for particular viewing directions or cross-track positions and likely due to
blocking objects in the optical path, started in June 2007 affecting a few positions. This effect



abruptly worsened in January 2009 affecting ~1/3 of the cross-track positions (Kroon et al., 2011).
The impacts of RA not only evolve with time but also vary over the duration of an orbit. Analysis
indicates that radiances in the UV1 channels (shorter than ~310 nm) used in our retrievals might
have been affected at all positions (Personal communication with S. Marchenko) and are not
adequately flagged for RA. Therefore, we need to evaluate the impacts of instrument degradation
and especially row anomaly on the temporal performance of our ozone profile product. Currently,
we are planning an update of the ozone profile algorithm to maintain the long-term consistency of
the product. The update will include empirical correction of systematic errors caused by the
instrument degradation and row anomaly as a function of time. Such correction also requires us to
evaluate the long-term retrieval quality of our product.
To understand retrieval quality and the resulting spatial and temporal performance of our OMI
product, we evaluate our data from October 2004 through December 2014 against available
ozonesonde and MLS observations, respectively, in two papers. This paper evaluates our ozone
product including both ozone profiles and stratospheric and tropospheric ozone columns using
ozonesonde observations with a focus on retrieval quality in the troposphere. More than 27,000
ozonesonde profiles from both regular ozonesonde stations and field campaigns are used in this
study to provide a comprehensive and global assessment of the long-term quality of our OMI ozone
product. This paper is followed by the validation against collocated MLS data with a focus on the
retrieval quality in the stratosphere (Huang et al., 2016), also submitted to this special issue).
This paper is organized as follows: Section 2 describes OMI retrievals and ozonesonde data. The
validation methodology is introduced in Section 3. Section 4 presents results, analysis and
discussions regarding the OMI and ozonesonde comparisons. Section 5 summarizes and concludes
this study.
**2  OMI and Ozonesonde Datasets**
**2.1 OMI and OMI Ozone Profile Retrievals**
OMI is a Dutch-Finnish built nadir-viewing pushbroom UV/visible instrument aboard the NASA
Earth Observing System (EOS) Aura satellite that was launched into a sun-synchronous orbit in
July 2004. It measures backscattered radiances in three channels covering the 270-500 nm





wavelength range (UV1: 270-310 nm, UV2: 310-365 nm, visible: 350-500 nm) at spectral
resolutions of 0.42-0.63 nm (Levelt et al., 2006).  Measurements across the track are binned to 60
positions for UV2 and visible channels, 30 positions for the UV1 channels due to the weaker
signals. This results in daily global coverage with a nadir spatial resolution of 13 km × 24 km
(along × across track) for UV2 and visible channels, and 13 km × 48 km for the UV1 channel.
The SAO OMI ozone profile algorithm was adapted from the GOME ozone profile algorithm (Liu
et al., 2005) to OMI and was initially described in detail in Liu et al. (2010b). Profiles of partial
ozone columns are retrieved at 24 layers, ~2.5 km for each layer, from the surface to ~60 km using
OMI radiance spectra in the spectral region 270-330 nm with the optimal estimation technique. In
addition to the OC, SOC and TOC can be directly derived from the retrieved ozone profile with
the use of tropopause from the daily National Center for Environmental Protection (NCEP)
reanalysis data. The retrievals are constrained with month- and latitude-dependent climatological
a priori profiles derived from 15-year ozonesonde and SAGE/MLS data (McPeters et al., 2007)
with considerations of OMI random-noise errors. OMI radiances are pre-calibrated based on two
days of average radiance differences in the tropics between OMI observations and simulations
with zonal mean MLS data for pressure less than 215 hPa and climatological ozone profile for
pressure greater than 215 hPa. This "soft calibration" varies with wavelength and cross-track
positions but does not depend on space and time.
The updated algorithm of our SAO OMI ozone product was briefly described in Kim et al. (2013).
The radiative transfer calculations have been improved through the convolution of simulated
radiance spectra at high resolutions rather than effective cross sections, which is done by
interpolation from calculation at selected wavelengths assisted by weighting function. In addition,
four spatial pixels along the track are coadded to speed up production processes at a nadir spatial
resolution of 52 km × 48 km. Meanwhile, minimum measurement errors of 0.4% and 0.2% are
imposed in the spectral ranges 270-300 nm and 300-330 nm, respectively, to stabilize the
retrievals.  The use of floor errors typically reduces the Degree of Freedom for Signals (DFS) and
increases retrieval errors. Compared to the initial retrievals, the average total, stratospheric, and
tropospheric DFS decrease by 0.49, 0.27, and 0.22, respectively, and the mean retrieval errors in
OC, SOC, and TOC increase by 0.6, 0.5, and 1.2 DU, respectively. The corresponding changes to
the retrievals are generally within retrieval uncertainties except for a systematic increase in



tropospheric ozone at SZA larger than ~75°, where the TOC increases to ~12 DU. Validation
against ozonesonde data indicates that this TOC increase at large SZA makes the retrieval worse.
Therefore retrieved tropospheric ozone at such large SZA should not be used, but the retrieved
total ozone still shows good quality (Bak et al., 2015).
For current products, retrievals contain ~5.5-7.4 DFS, with 4.6-7.3 in the stratosphere and 0-1.2 in
the troposphere. Vertical resolution varies generally from 7–11 km in the stratosphere to 10–14
km in the troposphere, when there is adequate retrieval sensitivity to the tropospheric ozone.
Retrieval random-noise errors (i.e., precisions) typically range from 0.6–2.5 % in the middle
stratosphere to approximately 12% in the lower stratosphere and troposphere. The solution errors,
dominated by smoothing errors, vary generally from 1-7% in the middle stratosphere to 7-38% in
the troposphere. The solution errors in the integrated OC, SOC, and TOC are typically in the few
DU range. Errors caused by the forward model and forward model parameter assumptions are
generally much smaller than the smoothing error (Liu et al., 2005). The main sources of these
errors include systematic errors in temperature and cloud-top pressure. Systematic measurement
errors are the most difficult to estimate, mostly due to lack of full understanding of the OMI
instrument calibration.
Certain cross track positions in OMI data have been affected by RA since June 2007 (Kroon et al.,
2011). Loose thermal insulating material in front of the instrument's entrance slit is believed to
block and scatter light, causing measurement error. The anomaly affects radiance measurements
at all wavelengths for specific cross-track viewing directions that are imaged to CCD rows.
Initially, the anomaly only affected a few rows. But since January 2009, the anomaly has spread
to other rows and shifted with time. The RA also shows slight differences among different spectral
channels, and varies during the duration of an orbit. Pixels affected by the RA are flagged in the
level 1b data. The science team suggested that they are not be used in research. For data before
2009, the RA flagging is not applied in the processing. Pixels seriously affected by RA will
typically show enhanced fitting residuals. The algorithm was updated to use RA flagging in the
UV1 channel and was used to process the data starting from 2009. If a pixel is flagged as a row
anomaly then it is subsequently not retrieved to speed up the processing except that the cross-track
position 24 is still retrieved due to reasonably good fitting.  It should be noted that the retrieval
quality of those non-flagged pixels may still be affected by the RA, because of the different RA





flagging in the UV1 and UV2, the lack of RA flagging before 2009and inadequacy of the RA
flagging.
To screen out OMI profiles for validation, we only use OMI ozone profiles meeting the following
criteria based on three filtering parameters: 1) nearly clear-sky scenes with effective cloud fraction
less than 0.3; 2) cross track positions between 4 and 27 due to relatively worse quality and much
larger footprint size for these greater off-nadir positions; 3) SZA should be less than 75° due to
very limited retrieval sensitivity to tropospheric ozone and the aforementioned positive biases. We
will use all OMI pixels of each filtering parameter when evaluating retrieval quality as a function
of that specific parameter. The fitting quality of each retrieval is shown in the fitting RMS (root
mean square of the fitting residuals relative to the assumed measurement errors). The mean fitting
RMS including both UV1 and UV2 channels has been increasing with time as shown in Figure 1.
This is primarily due to the increase of fitting residuals in UV1 caused by the instrument
degradation and RA since the fitting residuals of UV2 only slightly increase with time. As
aforementioned, the retrieval information of stratospheric and tropospheric ozone mainly comes
from UV1 and UV2, respectively. Consequently, retrievals in the troposphere, the focus of this
paper, are less impacted by the increasing fitting RMS. However, to apply consistent filtering in
validation against both ozonesonde in this study and MLS data in the companion paper (Huang et
al., 2016, submitted to the same special issue), we set the RMS threshold based on the overall
fitting RMS and select retrievals with fitting RMS smaller than the sum of monthly mean RMS
and its 2σ (i.e., Standard Deviations (SDs) of fitting RMS).
**2.2  Ozonesondes**
The balloon-borne ozonesonde is a well-established technique to observe the ozone profile from
the surface to ~35 km with vertical resolution of ~100-150 m and approximately 3-5% precision
and 5-10% accuracy (Komhyr, 1986; Komhyr et al., 1995; Johnson, 2002; Smit et al., 2007;
Deshler et al., 2008). Ozonesonde data have been widely used in the studies of stratospheric ozone,
climate change, tropospheric ozone and air quality, as well as the validation of satellite
observations (Kivi et al., 2007; Wang et al., 2011; Huang et al., 2015; Thompson et al., 2015).
However, the accuracy of ozonesonde observations depends on data processing technique, sensor
solution, and instrument type and other factors. Consequently, station-to-station biases may occur



in ozonesonde measurements and could be as great as 10% (Thompson et al., 2007c; Worden et
al., 2007).
A decade (2004-2014) of global ozonesonde data with locations shown in Figure 2, are utilized in
this study to validate our OMI ozone profile product. Most of our ozonesonde data were obtained
from the Aura Validation Data Center (AVDC) archive. It contains routine launches from
ozonesonde stations, mostly weekly and occasionally 2-3 times a week at some stations. It also
collects launches from field campaigns, for instance, IONS 06 (INTEX-B Ozone Network Study
2006), ARCIONS (Arctic Intensive Ozonesonde Network Study)
(http://croc.gsfc.nasa.gov/arcions/) (Thompson et al., 2008; Tarasick et al., 2010). Data not
available at AVDC are obtained from other archives such as the World Ozone and Ultraviolet
Radiation Data Center (WOUDC) (http://woudc.org/), the Southern Hemisphere Additional
Ozonesondes (SHADOZ) (Thompson et al., 2007a; Thompson et al., 2007b), as well as archives
of recent field campaigns including DISCOVER-AQ (Deriving Information on Surface Conditions
from Column and Vertically Resolved Observations Relevant to Air Quality, http://discover-
aq.larc.nasa.gov/) (Thompson et al., 2015) and SEACR$^4$S (Studies of Emissions and Atmospheric
Composition, Clouds and Climate Coupling by Regional Surveys,
https://espo.nasa.gov/home/seac4rs) (Toon et al., 2016). Almost all of the ozonesonde data in this
study were obtained from electrochemical concentration cell (ECC) ozonesondes, which is based
on the oxidation reaction of ozone with potassium iodide (KI) in solution. The exceptions are
Hohenpeissenberg station in Germany that uses Brewer-Mast (BM) ozonesondes, the New Delhi,
Poona, and Trivandrum stations that use Indian ozonesondes, and four Japanese stations (i.e.,
Sapporo, Tsukuba, Naha and Syowa) that switched from KC ozonesondes to ECC ozonesondes
during late 2008 and early 2010. These types of ozonesondes have been reported to have larger
uncertainties than ECC ozonesondes (WMO, 1998; Liu et al., 2013; Hassler et al., 2014).
To avoid using anomalous profiles, we screen out ozonesondes that burst at pressure exceeding
200 hPa, ozone profiles with gaps greater than 3 km, more than 80 DU TOC or less than 100 DU
SOC. In the SOC comparison, we also filter measurements that do not reach 12 hPa. Some
ozonesonde data used in this paper (e.g. WOUDC data) are provided with a correction factor (CF)
derived by normalizing the integrated ozone column (appended with ozone climatology above





burst altitude) to the coincident total ozone column measured by a Dobson or Brewer instrument
to account for uncertainties in pump efficiency especially near the top of the profiles. The CF is
also included in our screening processes. If the CF is available, we select ozonesonde profiles with
the CF in the range of 0.85 to 1.15 to filter profiles that require too much correction, and apply the
correction. Finally, a small number of obviously erroneous profiles are visually examined and
rejected.
**3   Comparison Methodology**
Previous studies on the validation of satellite observations used a range of coincidence criteria.
Wang et al. (2011) set a 100 km radius and 3 hour time difference as coincidence criteria. Kroon
et al. (2011) applied coincidence criteria of $\pm 0.5°$ for both latitude and longitude and 12 hours.
In this paper, we determine our coincident criteria based on the balance between finding most
coincident OMI/ozonesonde pairs to minimize differences due to spatiotemporal samplings and
finding a sufficient number of pairs for statistical analysis. For each screened ozonesonde profile,
we first select all filtered OMI data within $\pm 1°$ latitude, $\pm 3°$ longitude and $\pm 6$ hours and then find
the nearest OMI retrieval within 100 km from the ozonesonde station to perform the validation on
the individual profile basis.
Ozonesondes have much finer vertical resolution than OMI retrievals. To account for the different
resolutions, ozonesonde profiles are first integrated into the corresponding OMI vertical grids and
then degraded to the OMI vertical resolution by using the OMI retrieval Averaging Kernels (AKs)
and *a priori* ozone profile based on the following equation:
$$\hat{x} = x_a + A(x - x_a), \tag{1}$$
where $x$ is the ozonesonde profile integrated into the OMI grid, $\hat{x}$ is the retrieved ozone profile if
the ozonesonde is observed by OMI, , $A$ is the OMI AK matrix, and $x_a$ is the OMI *a priori* ozone
profile. We refer to this retrieval as "convolved ozonesonde profile", which is a reconstruction of
ozonesonde profile with OMI retrieval vertical resolution and sensitivity. Missing ozone profiles
above ozonesonde burst altitude are filled with OMI retrievals. The convolution process essentially
removes OMI smoothing errors and the impacts of a priori from the comparison so that
OMI/ozonesonde differences are mainly due to OMI/ozonesonde measurement precision,





spatiotemporal sampling differences and other errors. However, in the regions and altitudes where
OMI has low retrieval sensitivity, the comparisons can show good agreement because both the
retrieval and convolved ozonesonde approach the a priori profile. To overcome the limitation of
such a comparison, we also compare with unconvolved ozonesonde profiles since it indicates how
well the retrievals can represent the actual ozonesonde observations (i.e., smoothing errors are
included as part of retrieval errors). In addition, we also compare OMI a priori and
convolved/unconvolved ozonesonde profiles to indicate the retrieval improvement over the a
priori.
For consistent calculations of TOC and SOC from the OMI/ozonesonde data, the tropopause
pressure included in the OMI retrieval and ozonesonde burst pressure (required to be less than 12
hPa or ~30 km) are used as the proper boundaries. The TOC is integrated from the surface to the
tropopause and the SOC is integrated from the tropopause pressure to the ozonesonde burst
pressure.
The relative profile difference is calculated as (OMI- Sonde) / OMI a priori ×100% in the present
comparison with ozonesonde and with MLS in the companion paper. Choosing OMI a priori rather
than MLS/ozonesonde is to avoid unrealistic statistics skewed by extremely small values in the
reference data especially in the MLS retrievals of upper troposphere and lower stratosphere ozone
(Liu et al., 2010a). Unlike the profile comparison, ozonesonde/OMI SOC/TOC values are used in
the denominator in the computation of relative difference. To exclude remaining extreme outliers
in the comparison statistics, values that are exceeding $3\sigma$ from the mean differences are filtered.
After applying the OMI/ozone filtering and coincident criteria, approximately 10,500 ozonesonde
profiles are used in the validation. We performed the comparison for five latitude bands: northern
high latitudes (60° N-90° N), northern mid-latitudes (30° N-60° N), tropics (30° S-30° N), southern
mid-latitudes (60° S-30° S), and southern high latitudes (90° S-60° S) to understand the latitudinal
variation of the retrieval performance.  We investigated the seasonal variations of the comparisons
mainly at northern mid-latitudes where ozone retrieval shows distinct seasonality and there are
adequate coincidence pairs. To investigate the RA impacts on OMI retrievals, we contrasted the
comparison before (2004-2008, i.e., pre-RA) and after (2009-2014, i.e., post-RA). Although we
filter OMI data based on cloud fraction, cross-track position, and SZA, we conduct the comparison



as a function of these parameters using coincidences at all latitude bands to show how these
parameters affect the retrieval quality. In these evaluations, the filtering of OMI data based on
cloud fraction, cross-track position, and SZA are switched off, respectively. Approximately 15,000
additional ozonesonde profiles are used in this extended evaluation. To evaluate the long-term
performance of our ozone profile retrievals, we analyze the monthly mean biases (MBs) of the
OMI/ozonesonde differences as a function of time using coincidences in the 60° S-60° N region
and then derive a linear trends over the entire period as well as the pre-RA and post-RA periods.
**4   Results and Discussions**
**4.1 Comparison of Ozonesonde and OMI profiles**
**4.1.1    Ozone Profile Differences**
Comparisons of ozone profiles between OMI/a priori and ozonesondes with and without applying
OMI AKs for the 10-year period (2004-2014) are shown in the left panels of Figure 3. The MBs
and SDs vary spatially with altitude and latitude. Vertically, the SD typically maximizes in the
upper troposphere and lower troposphere (UTLS) in all latitude bands due to significant ozone
variability and a priori uncertainty. Bak et al. (2013b) showed that the use of Tropopause-Based
(TB) ozone profile climatology with NCEP Global Forecast System (GFS) daily tropopause
pressure can significantly improve the a priori, and eventually reduce the retrieval uncertainty.
Consequently, the SDs of OMI/sonde differences in the UTLS at mid- and high-latitudes can be
reduced through reducing the retrieval uncertainties. Latitudinally, the agreement is better in the
tropics and becomes worse at higher latitudes. The patterns are generally similar in the northern
and southern hemispheres. The MBs between OMI and ozonesonde are within ~6% with AKs and
10% without AKs in the tropics and the middle latitudes. Large changes in the biases between with
and without AKs occur in the tropical troposphere where the bias differences reach 10%. The MBs
increase to 20-30% at high latitudes consistently with large oscillation from ~-20-30% at ~300 hPa
to +20% near the surface both with and without the application of AKs. At pressure < 50 hPa, the
SDs for comparisons with OMI AKs are typically 5-10% at all latitudes except for the 90° S-60°
S region. For pressure > 50 hPa, the SDs are within 18% and 27% in the tropics and middle-



latitudes, respectively, but increase to 40% at higher latitudes. The SDs for comparison without
applying OMI AKs, i.e., including OMI smoothing errors in the OMI/ozonesonde differences,
typically increase up to 5% for pressure < 50 hPa, but increase up to 15-20% for pressure > ~50hPa.
The smoothing errors derived from root square differences of the MBs with and without OMI AKs
are generally consistent with the retrieval estimate from the optimal estimation.
The improvements of OMI over the climatological (a priori) profiles can be reflected in the
reduction of MBs and SDs in the comparisons between ozonesondes and OMI retrievals, and
between ozonesondes and a priori. The retrieval improvements in the MBs are clearly shown in
the tropics and at ~ 100 hPa pressure in the middle latitudes. At high latitudes, the MBs and
corresponding oscillations in the troposphere are much larger than these in the a priori comparison,
suggesting that these large biases are mainly caused by other systematic measurements errors at
high latitudes (larger SZAs and thus weaker signals).  As can be seen from the reduction of SDs,
OMI retrievals show clear improvements over the a priori at pressure < 300 hPa. For pressure >
300 hPa, the retrieval improvements vary with latitudes. There are consistent retrieval
improvements throughout the surface - 300hPa layer in the tropics and only the 550 - 300 hPa
layer at middle latitude, while there is no retrieval improvement over the a priori for > 300 hPa at
high latitudes. The failure to improve the retrieval over a priori in part of the troposphere at middle
and high latitudes is caused by several factors. They are the inherent reduction in retrieval
sensitivity to lower altitudes at larger SZAs as a result of reduced photon penetration into the
atmosphere, unrealized retrieval sensitivity arising from retrieval interferences with other
parameters (e.g., surface albedo) as discussed in Liu et al. (2010b) and the use of floor-noise of
0.2% that underestimate the actual OMI measurement SNR. In addition, the a priori ozone error
in the climatology is quite small since the SDs of the differences between the a priori and
ozonesonde without AKs are typically less than 20% in the lower troposphere for middle and high
latitudes, which also makes it more difficult to improve over the a priori comparison.
The right column of Figure 3 shows the comparisons between OMI retrievals and ozonesondes
convolved with OMI AKs in the pre-RA and post-RA periods, respectively. In the tropics and mid-
latitudes, the pre-RA comparison is better than the post-RA comparison, with SDs smaller by up
to ~8% at most altitudes especially in the troposphere. The pre-RA comparison also shows smaller
biases near ~300 hPa at middle latitudes while the post-RA comparison exhibits negative biases



reaching 8-12%. At high latitudes, the pre-RA period does not show persistent improvement
during the post-RA period. The pre-RA comparison shows slightly smaller SDs at most altitudes
and smaller negative biases by 10% around 300 hPa in the northern high latitudes, and smaller
positive biases by 20% near the surface in the southern high latitudes. The worse results during
the post-RA period are caused by increasingly noisy OMI measurements with smaller SNR and
the additional radiometric biases made by the RA, which vary with space and time. The smaller
SDs at some altitudes of high latitudes may reflect a combination of ozone variation, uneven
distribution of ozonesondes with varying uncertainty at different stations, and cancellation of
radiometric errors by the RA.
As seen from the number of OMI/ozonesonde coincidences shown in Figure 3, the northern mid-
latitudes and the tropics have sufficient coincidences to validate the retrievals as a function of
season. In the tropics, the retrieval comparison does exhibit little seasonality as expected (not
shown). Figure 4 shows the comparison similar to Figure 3(c) for each individual season at
northern middle latitudes. The comparison results are clearly season-dependent with best
agreement in the summer (except for the MBs) and the worst agreement in the winter. This
indicates the general best retrieval sensitivity to lower tropospheric ozone during the summer as a
result of small SZAs and stronger signals and worst retrieval sensitivity during the winter as a
result of large SZAs and weaker signals. The MBs for with and without AKs at 300 hPa vary from
~12% in the winter to -10% in the summer. The overall MBs are the smallest during the spring,
within 6%; but the MBs at pressure < 50 hPa are the best during the summer. The maximum SDs
vary from 31% in the winter to 20% in the summer. Also, the retrieval in the summer shows the
most improvements over the a priori in the lower troposphere at all tropospheric layers except for
the bottom layer, while the retrievals during other seasons show the improvement over a priori
only above the lowermost two/three layers. The seasonal variation of retrieval quality is partially
caused by the seasonal variations of the retrieval sensitivity and ozone variability. Bak et al.
(2013b) showed that the use of TB ozone climatology with daily NCEP GFS tropopause pressure
can significantly reduce the seasonal dependence of the comparison with ozonesondes. In addition,
radiometric calibration errors such as those caused by stray light and RA also contribute to the
seasonal variation of retrieval quality.





### 4.1.2 Solar Zenith Angle Dependence

The SZA of low earth orbit (LEO) satellite observation varies latitudinally and seasonally; therefore the SZA dependence of the retrieval can cause latitudinal and seasonal dependent retrieval biases. SZA is one of the main drivers that affect retrieval sensitivity especially to tropospheric ozone. At large SZA, the measured backscattered signal becomes weak due to weak incoming signal and long path length; the retrieval sensitivity to the tropospheric ozone decreases due to reduced photon penetration to the troposphere. In addition, measurements are subject to relatively larger radiometric errors such as those from stray light and as a result of weaker signal, and radiative transfer calculations can lose accuracy at larger SZA (Caudill et al., 1997).

Figure 5 gives the MBs and SDs of differences between OMI and ozonesondes (with OMI AKs) in a function of SZAs. We can see that retrieval performance generally becomes worse at large SZA. The SD typically increases with SZA especially at pressure > 300 hPa. At SZA larger than 75°, the SD at ~300 hPa increases to greater than ~45%. The variation of MBs with SZA is more complicated. We see generally larger positive biases at larger SZA in the troposphere with > 20% biases at SZA larger than 75°. The MBs near ~ 30 hPa becomes more negative at larger SZAs. There is a strip of positive biases of ~10% that slightly decreases in pressure from ~50 hPa at low SZA to ~10 hPa at large SZA; it might be due to some systematic radiometric biases that can affect ozone at different altitudes varying with SZA. Because of the clear degradation of the retrieval quality at large SZA, we set the SZA filtering threshold of 75° to filter OMI data.

### 4.1.3 Cloud Fraction Dependence

The presence of cloud affects retrieval sensitivity since clouds typically reduce sensitivity to ozone below clouds and increase sensitivity to ozone above clouds. The accuracy of ozone retrievals is sensitive to the uncertainties of cloud information and cloud treatment (Liu et al., 2010a; Antón and Loyola, 2011; Bak et al., 2015). Our OMI ozone algorithm assumes clouds as Lambertian surfaces with optical centroid cloud pressure, and partial clouds are modeled using independent pixel approximation such that the overall radiance is the sum of clear and cloudy radiances weighted by the effective cloud fraction. The cloud albedo is assumed to be 80% and is allowed to vary (>80%) with the effective cloud fraction.



Figure 6 gives the influences of effective cloud fraction on the comparisons between OMI and
ozonesonde observations convolved with OMI AKs. The MBs and SDs do not change much with
cloud fraction for pressure < 100 hPa, and typically increase with the increase of cloud fraction for
pressure > 100 hPa. The MBs at pressure > 100 hPa, especially greater~300 hPa, increase to more
than 10% with cloud fraction greater than ~0.3. This indicates that the cloud fractions have small
impacts on the stratospheric retrievals but large impacts on the tropospheric retrievals as expected.
Some of the variation with cloud fraction such as negative biases near ~300 hPa at cloud fraction
of ~0.4 and the decreases of positive biases at ~ 50 hPa for cloud fraction greater than ~0.8 may
be partially related to the uncertainties of the cloud parameters. The chosen filtering threshold of
0.3 in cloud fraction is a tradeoff between validating OMI data with adequate retrieval sensitivity
to tropospheric ozone and finding adequate number of OMI/ozonesonde coincidences.

### 4.1.4    Cross-Track Position Dependence

The OMI swath is divided into 30 cross-track pixels at the UV1 spatial resolution of our product.
Each cross-track position is measured by a different part of the CCD detector, i.e., essentially a
different instrument. Radiometric calibration coefficients of the instrument are characterized
during pre-launch only at selected CCD column pixels and then interpolated to other columns,
causing variation in the radiometric calibration performance across the CCD detector. This in turn
causes cross-track dependent biases in the calibrated radiance (Liu et al., 2010b), which therefore
causes stripping in almost all the OMI data products if no de-striping procedure is applied. Our
retrieval algorithm has included a first-order empirical correction independent of space and time
to remove the cross-track variability (Liu et al., 2010b). However, residual dependence on cross-
track position remains and the radiometric calibration at different position can degrade differently
with time (e.g., the RA impact). In addition, the viewing zenith angle ranges from ~0° to ~70° and
the footprint area increases by approximately an order of magnitude from nadir to the first/last
position. So the varying viewing zenith angle causes the variation of retrieval sensitivities and
atmospheric variabilities within varying footprint areas may also cause additional cross-track
dependence in the retrieval performance.
Figure 7 provides the MBs and SDs of the differences between OMI and ozonesonde convolved
with OMI AKs as a function of cross-track position for pre-RA and post-RA periods, respectively.



It clearly exhibits cross-track dependence especially with large positive/negative MBs and large
SDs at the first/last several extreme off-nadir positions. This is why we select cross-track positions
of 4-27 in the validation to avoid positions with large biases. The enhanced biases/SDs at positions
24 (RA flagging not applied) and 27 (flagged as RA in UV2 since June 25, 2007 but not
flagged/applied in UV1) are due to the RA impact during the post-RA period. Cross-track positions
1-10 show consistent bias patterns with negative biases in ~300- 50 hPa layer and positive biases
in ~surface – 300 hPa layer, and large standard deviation around ~ 300 hPa although the magnitude
decreases with increasing cross-track position. This pattern occurs during both pre-RA and post-
RA periods although the values are larger during the post-RA period. For other cross-track
positions, the variation is relatively smaller but we can still see small striping patterns.
**4.2 Comparison of Partial Ozone Columns**
We investigate and validate OMI partial ozone columns, including SOCs, TOCs, and surface-550
hPa and surface-750 hPa ozone columns in this section. We define the lowermost one and two
layer as surface-750 hPa and surface-550 hPa in this paper, respectively, for conveniences.
Similarly, we also analyze the validation results of SOCs and TOCs during pre-RA and post-RA,
respectively, to test the impacts of RA on OMI partial ozone columns. In addition, we validate
ozone columns from the surface to ~550 hPa (bottom two layers) and ~ 750 hPa (bottom one layer)
against ozonesonde observations in the tropics and mid-latitude summer where there is better
retrieval sensitivity to these quantities.
**4.2.1    Comparison of Stratospheric Ozone Columns (SOCs)**
The left column of Figure 8 shows the MBs and SDs of the comparisons of OMI and ozonesonde
SOCs for each of the five latitude bands during 2004-2014. In all regions, the OMI SOCs have
excellent agreement with ozonesonde SOCs regardless of whether ozonesonde data are convolved
with OMI AKs. The application of OMI AKs to ozonesonde SOCs only slightly improves the
comparison statistics. The MBs with OMI AKs are within 1.8% except for a negative bias of 3%
at northern high latitudes, while the SDs are within 5.1% except for 5.7% at high latitudes. The
correlation coefficient is greater than 0.95 except for 0.90 in the tropics due to the smaller SOC
range. The SDs are typically larger than the comparisons with MLS data (Liu et al., 2010a) due to





worse coincidence criteria, relatively larger uncertainty in the ozonesonde stratospheric ozone
columns compared to MLS data, and different altitude ranges of integration.
The middle and right columns of Figure 8 show comparison results during the pre-RA and post-
RA periods, respectively. The comparison is typically better during the pre-RA with SDs smaller
by 0.2-0.6% and larger correlation coefficients although the MBs are generally smaller during the
post-RA period. One exception is at southern high-latitudes where the post-RA comparison
statistics are significantly better except for the MB, consistent with Figure 3, likely due to a
combination of ozone variation between these two periods, uneven distribution of ozonesondes at
different stations, and cancellation of various calibration errors.

### 510     4.2.2    Comparison of Partial Ozone Columns in the Troposphere

The left column of Figure 9 shows the comparison of OMI and ozonesonde (with and without OMI
AKs) TOCs for each of the five latitude bands during 2004-2014. Without applying OMI AKs, the
MBs are within 1-3% except for 9% at northern high latitudes; The SDs are within 20% in the
tropics and mid-latitudes and increase to ~30-40% at high-latitudes. The correlation coefficient
ranges from 0.83 in the tropics to ~0.7 at middle latitudes, and 0.5-0.6 at high-latitudes. The linear
regression slopes are in the range 0.6-0.8 typically smaller at high latitudes due to reduced retrieval
sensitivity to the lower troposphere. After applying the OMI AKs to ozonesonde data to remove
smoothing errors, we see significant improvement in the comparison statistics except for MBs,
which are within 6% at all latitudes. The SDs are reduced to within 15% in the tropics and middle
latitudes and ~30% (5.5-8.1 DU) at high latitudes; the correlation improves by 0.04-0.12 and  the
slope significantly increases by 0.12-0.23 to the range 0.8-1.0 at different latitude bands due to
accounting for inadequate retrieval sensitivity to the lower and middle troposphere.
The middle and right columns of Figure 9 show comparisons during pre-RA and post-RA,
respectively. The comparison between OMI and ozonesondes with OMI AKs TOCs during the
pre-RA period is significantly better than these during the post-RA period in the tropics and mid-
latitudes with SDs smaller by 3.4-5.5% and greater correlation. The MBs during the post-RA
period is smaller by ~2 DU at mid-latitudes, but larger by ~1 DU in the tropics. However, the post-
RA comparison is similar to the pre-RA comparison at northern high-latitudes and is even better
at southern high latitudes probably due to the aforementioned ozonesonde issues.





Figure 10 shows examples of time series when comparing individual OMI and ozonesondes (with
OMI AKs) TOCs and their corresponding differences at six selected stations, one for each latitude
region of 90° N-60° N, 60° N-30° N, 30° N-0°, 0°-30° S, 30° S-60° S and 60° S-90° S. OMI TOC
shows good agreement with ozonesondes at these stations with overall MBs ≤ 3 DU and SDs less
than 5.1 DU. The comparison is also good even in the high latitude regions partially because the
Summit and Neymayer stations only have ozonesonde launches during local summer. Seasonal
dependent biases are clearly seen at Payerne, and bias trends can be seen at several stations with
positive trends at Summit and Neumayer and a negative trend at Naha. In the pre-RA and post-RA
periods, the MBs are typically within 2 DU and the SDs are typically smaller during the pre-RA
period except for Naha. The better comparison (both mean bias and standard deviation) during the
post-RA period at Naha is likely due to the switch to ECC ozonesondes beginning on November
13, 2008 from KC ozonesonde that have greater uncertainty (WMO, 1998).
Figure 2 also shows the MBs and SDs of the TOC differences between OMI and ozonesonde
convolved with OMI AKs at each station/location where there are at least 10 coincident
OMI/ozonesonde pairs. OMI data generally exhibit good agreement with ozonesondes at most of
the stations, with MBs of ≤ 3 DU and SDs of ≤ 6 DU. In the tropics (30° S-30° N), very large SDs
(>11 DU) occur at the two Indian stations (New Delhi, and Trivandrum). The large bias of >6 DU
at New Delhi is likely associated with the large uncertainties of the Indian ozonesonde data. Hilo
has large biases of ~4.5 DU with 3.2 and 6.2 DU for pre-RA and post-RA, respectively. Java also
has a large bias of ~5 DU but shows no much difference between pre-RA and post-RA. Consistent
~2% and ~5% underestimates of OC by ozonesondes compared to OMI total ozone are found in
Hilo and Java, respectively (Thompson et al., 2012). These OC underestimates may partly explain
the large TOC biases in Hilo and Java. However, the reason for underestimates of ozonesonde-
derived OC is unknown. In the middle latitudes, noticeably large SDs and/or biases occur at a few
stations such as Churchill, Sable Islands, Hohenpeissenberg, and Parah. Three Japanese stations,
Sapporo, Tateno, and Naha, exhibit relatively large biases of 2-3 DU and even larger biases before
switching from KC to ECC sondes. Almost half of the 11 northern high latitude stations (60° N-
90° N) and two of the 6 southern high-latitude stations have large SDs/biases. In addition to
retrieval biases from the OMI data, some of the large biases or SDs might be partially related to
ozonesonde type with different biases and uncertainties due to different types (e.g., Indian sonde





stations, Brewer-Mast ozonesonde at Hohenpeissenberg, three KC sonde stations), manufacturers
(e.g., SP vs. ENSCI for ECC sonde), sensor solution or related to individual sonde operations,
which was shown in the validation of GOME ozone profile retrievals (Liu et al., 2006a).
Figure 11 shows the comparison for each season at northern mid-latitudes. Consistent with profile
comparison, the TOC comparison is season-dependent. When applying OMI AKs, the mean bias
varies from 3 DU in winter to -1.5 DU in summer. The SDs are within 6.8 DU with the smallest
value during fall due to less ozone variability. The regression slopes are very close, within 0.04
around 0.67. The retrieval sensitivity is smallest during the summer as seen from the greatest
correlation and slope and relatively small standard deviation, and is the worst during the winter.
With OMI AKs applied to ozonesonde profiles, the MBs only slightly change (varying from 3.5
DU to -1.3 DU), but the SDs are significantly reduced to within 5.2 DU, the slopes significantly
increase by ~0.2 to 0.8-1.0, and the correlation improves significantly during the winter and spring.
Figure 12 compares the surface~550 hPa and surface~750 hPa ozone columns with ozonesonde
data in the middle latitudes during summer and the tropics. Compared to the TOC comparisons in
Figure 9 and Figure 11, the comparisons of these lower tropospheric ozone columns exhibit smaller
regression slopes and correlations that are a result of reduced retrieval sensitivity. In the tropics,
the slopes decrease from 0.78 in TOC to 0.65 in the surface~550 hPa ozone column and ~0.50 in
the surface~750 hPa column, with corresponding correlation from 0.83 to 0.74 in the surface-~550
hPa column, and 0.66 in the surface-~750 hPa column. This indicates that the retrievals in the
surface~550 hPa/750 hPa can capture ~65%/50% of the actual ozone change from the a priori.
During the middle latitude summer, the slope decreases from 0.71 in the TOC comparisons to 0.42
in the surface-~550 hPa comparisons and 0.32 in the surface-~750 hPa comparisons, with
corresponding correlation coefficients from 0.74 to 0.5 and 0.46. Thus, the retrievals in the
surface~550 hPa and ~750 hPa only capture ~40%/30% of the actual ozone change from the a
priori. The MBs are generally small within 0.5 DU (5%) with SDs of ~3.6 DU (20-28%) in the
surface~550 hPa ozone column and ~2.5 DU (25-36%) in the surface~750 hPa ozone column.
After applying OMI AKs to account for inadequate retrieval sensitivity and removing smoothing
errors, the slope significantly increases to approach 1 (as expected).  SDs are reduced to ~10% in
the middle latitudes and ~15% in the tropics.



### 4.3 Evaluation of Long-term Performance

Previous evaluation indicated systematic differences between pre-RA and post-RA periods and generally worse performance during the post-RA periods. To further illustrate the long-term stability of our ozone profile product and understand the quality of OMI radiometric calibration as a function of time, we analyze monthly MBs of OMI/ozonesonde differences with OMI retrieval AKs in ozone profiles, SOCs, and TOCs. Due to the lack of OMI observations during some months at high-latitudes, we focus the evaluation by using coincidence pairs in 60° S-60° N. Monthly MBs are calculated only if there are more than 5 OMI-ozonesonde pairs in a given month. Linear regression trend is on the MBs for the entire period (2004-2014) and/or for the pre-RA and post-RA periods, respectively. The trend is considered statistically significant if its P value is less than 0.05.

The linear trends of monthly mean ozone biases for each OMI layer between 60° S-60° N are plotted in Figure 13 for each of the three periods. During 2004-2014, marked in black, ozone biases at layers above 50.25 hPa show significant positive trends of 0.06-0.17 DU/year (0.17-0.52%/year), while ozone biases between 290 hPa and 110 hPa exhibit significant negative trends of 0.1-0.19 DU/year (1-2%/year). The positive trends in the stratosphere are generally consistent with those shown in OMI-MLS comparisons (Huang et al., 2016). In the lowermost three OMI layers, ozone differences are more stable but with several large spikes during the post-RA periods likely due to the RA evolution or instrument operation. The derived trends for the pre-RA period are generally more flat and insignificant at all layers indicating good stability of our product as well as the OMI radiometric calibration. During the post-RA period, the derived trends are positive above 75 hPa with statistical significance. These positive trends in the stratosphere are generally similar to those over the entire period, suggesting the dominant contribution of the post-RA period to the overall trend. In the altitude range 214 – 108 hPa, the post-RA trends are also flat similar to the pre-RA trends, but the values are systematically smaller during the post-RA period, causing significantly negative trends over the entire period.

The SOC biases exhibit small positive trend of 0.14±0.09 DU/year in 2004-2014 with no statistical significance (Figure 14(a)). This slight positive trend is a result of trend cancellation by the positive trends above 80 hPa and negative trends between 220 hPa and 80 hPa The TOC biases reveal a



significant negative trend of -0.18 ± 0.05 DU/year (Figure 14(b)), mostly from layers in the upper
troposphere. In the pre-RA and post-RA periods, both trends of both SOC and TOC biases are
relatively flat during the pre-RA period, while the SOC trend in the post-RA period is 0.77 ± 0.20
DU/year with significance. It is noticeable that the P value of TOC trend in the post-RA period is

622    0.06.

The significant trends of ozone biases at different layers as well as in SOC and TOC suggest that
the current ozone profile product is not suitable for trend studies especially during the post-RA
period. The relatively flat bias trends during the pre-RA periods and statistically significant trends
during the post-RA period confirm that the better stability of our product during the pre-RA period
and more temporal variation of the retrieval performance during the post-RA period are likely
associated with the RA evolution. In previous sections, the validation of our retrievals revealed
latitudinal/seasonal/SZA and cross-track dependent biases even during the pre-RA period. This
indicates the need to remove signal dependent errors and the calibration inconsistency across the
track. To maintain the spatial consistency and long-term stability of our ozone profile product, we
need to further improve OMI's radiometric calibration especially during the post-RA period.
Preferably, the calibration improvement should be done in the level 0-1b processing. If this option
is not possible, we can perform soft calibration similar to Liu et al. (2010b) but derive the
correction as a function of time and latitude/SZA. In addition, it should be noted that the trend
calculation might be affected by factors such as the availability of correction factors with
ozonesondes (Morris et al., 2013), station-to-station variability and the uneven spatiotemporal
distribution of the ozonesondes, which can introduce considerable sampling biases (Liu et al.,
2009; Saunois et al., 2012).
**5   Summary and Conclusion**
We conducted a comprehensive evaluation of the quality of OMI ozone profile (PROFOZ)
products produced by the SAO algorithm, including their spatial consistency and long-term
performance using coincident global ozonesonde observations during the decade 2004-2014. To
better understand retrieval errors and sensitivity, we compared the retrieved ozone profiles and a
priori profile at individual layers with ozonesondes before and after being degraded to the OMI
vertical resolution with OMI retrieval average kernels (AKs). We also compared the integrated



SOC, TOC, and surface-~550/~750 hPa ozone columns with ozonesonde data. To understand the
spatial distribution of retrieval performance, the validations are grouped into five latitude ranges:
northern/southern high/middle latitudes, and the tropics. To investigate the impacts of the OMI
row anomaly (RA) on the retrievals, we contrasted the comparison before and after the occurrence
of major OMI RA in January 2009, i.e., pre-RA (2004-2008) and post-RA (2009-2014) periods.
In addition, we quantified the dependence of retrieval performance on seasonality and several key
parameters including solar zenith angle (SZA), cloud fraction, and cross-track position. Finally,
we analyzed the monthly mean variation of the mean biases (MBs) to examine the long-term
stability of the PROFOZ product.
The comparison between OMI and ozonesonde profiles varies in altitude, with maximum standard
deviations (SDs) in the Upper Troposphere and Lower Stratosphere (UTLS) due to significant
ozone variability, and varies with latitude similarly in the northern and southern hemispheres.
There is good agreement throughout the atmosphere in the tropics and mid-latitudes. With the
application of OMI AKs to ozonesonde data, the MBs are within 6%, and the SDs increase from
5-10% for pressure < ~50 hPa to within 18%(27%) in the tropics/mid-latitudes for pressure > ~50
hPa. In the high latitudes, the retrievals agree well with ozonesondes only for pressure < ~50 hPa
with MBs of < 10% and SDs of 5-15% for pressure < ~ 50 hPa, but with MBs reaching 30% and
SDs reaching 40% for pressure > ~50 hPa. The comparison is seasonally dependent. At northern
mid-latitudes, the agreement is generally best (except for MBs) in the summer, with the best
retrieval sensitivity and the smallest SDs as great as 20%, and the worst in the winter with the
worst retrieval sensitivity and the largest SDs reaching 31%. The MBs near 300 hPa vary from
12% in the winter to -10% in the summer. The post-RA comparison is generally worse in the
tropics and mid-latitudes than the pre-RA comparison, with SDs larger by up to 8% in the
troposphere and 2% in the stratosphere, and with larger MBs around ~300 hPa in the mid-latitudes.
But at high latitudes, the pre-RA comparison does not show persistent improvement over the post-
RA comparison, with smaller biases and larger SDs at some altitudes, especially at southern high
latitudes. The retrieval improvement over a priori can be determined from the SD reduction of the
retrieval comparison from the a priori comparison. The retrievals demonstrate clear improvement
over the a priori down to the surface in the tropics, but only down to ~750 hPa during mid-latitude
summer, ~550 hPa during the other seasons of mid-latitudes and ~ 300 hPa at high latitudes.



Retrieval performance typically becomes worse at large SZA, especially at SZA larger than 75°,
where the MBs in the troposphere are >20% and the SDs near ~300 hPa are > 45%. The worse
performance at larger SZA is due to a combination of weaker signal and greater influence by
radiometric calibration errors such as due to stray light, and radiative transfer calculation errors.
The variation of SZA is likely responsible for the majority of the retrieval dependence on latitude
and season. The retrieval quality for pressure > ~100 hPa degrades with increasing cloudiness in
terms of MBs and SDs, with MBs greater than 10% at cloud fraction > 0.3. The retrieval
performance also varies with cross-track position, especially with large MBs and SDs at the
first/last extreme off-nadir positions (e.g., 1-3 and 28-30). The dependence is stronger during the
post-RA period.
The integrated SOCs and TOCs also exhibit good agreement with ozonesondes. With the
convolution of OMI AKs to ozonesonde data, the SOC MBs are within 2% with SDs within ~5.1%
in the tropics and mid-latitudes. These statistics do not change much even without the applications
of OMI AKs. The comparison becomes slightly worse at high latitudes, with MBs up to 3% and
SDs up to 6%. The pre-RA comparison is generally better with smaller SDs of 0.2-0.6% except
for southern high latitudes, although with slightly larger MBs. The TOC MBs and SDs with OMI
AKs are within 6%, with SDs of <~15% in the tropics and mid-latitudes but reach 30% at high
latitudes. The pre-RA TOC comparison is also better in the tropics and mid-latitudes with SDs
smaller by 3.4-5.5% but worse values at southern high latitudes. The TOC comparison at northern
mid-latitudes varies with season, with MBs of 11%. There are worse correlation during winter
and MBs of -3% and best correlation in summer. The TOC comparison also shows noticeable
station-to-station variability in similar latitude ranges with much larger MBs and/or SDs at the two
Indian stations and larger MBs at several Japanese stations before they switched from KC
ozonesondes to ECC ozonesondes. This demonstrates the impacts of ozonesonde uncertainties
due to sonde types, manufacturers, sensor solution and operations. Without applying OMI AKs,
the TOC correlation with ozonesondes typically becomes worse at higher latitudes, ranging from
0.83 in the tropics to 0.5-0.6 at high latitudes. The linear regression slope is within 0.6-0.8,
typically smaller at higher latitudes, reflecting the smaller retrieval sensitivity down to the
troposphere at higher latitudes mainly resulting from larger SZA. The convolution of AKs





significantly improves the correlation and slope. The impact of retrieval sensitivity related to SZA
is also reflected in the seasonal dependence of the comparison at mid-latitudes.
The surface-~550/750 hPa ozone columns in the tropics during mid-latitude summer compare
quite well with ozonesonde data, with MBs of < 5% and SDs of 20-25%/28-36% without OMI
AKs. The correlation and slope decrease with decreasing altitude range due to reduced retrieval
sensitivity down to the lower troposphere. These columns capture ~65%/50% of the actual ozone
change in the tropics and ~40%/30% in the troposphere. Convolving ozonesonde data with OMI
AKs significantly increases the slope to ~1 and reduce the SDs to 10-15%.
The contrast of pre-RA and post-RA comparisons indicates generally worse post-RA performance
with larger SDs. Linear trend analysis of the OMI/ozonesonde monthly MBs further reveals
additional RA impact. The temporal performance over 60° S-60° N is generally stable with no
statistically significant trend during the pre-RA period, but displays a statistically significant trend
of 0.14-0.7%/year at individual layers for pressure < ~80 hPa, 0.7 DU/year in SOC and -0.33
DU/year in TOC during the post-RA period. Because of these artificial trends in our product, we
caution against using our product for ozone trend studies.
This validation study demonstrates generally good retrieval performance of our ozone profile
product especially in the tropics and mid-latitudes during the pre-RA period. However, the
spatiotemporal variation of retrieval performance suggests that OMI's radiometric calibration
should be improved, especially during the post-RA period, including the removal of signal-
dependent errors, calibration inconsistency across the track and with time to maintain the long-
term stability and spatial consistency of our ozone profile product.

**Acknowledgements**
This study was supported by the NASA Atmospheric Composition: Aura Science Team
(NNX14AF16G) and the Smithsonian Institution. The Dutch-Finnish OMI instrument is part of
the NASA EOS Aura satellite payload. The OMI Project is managed by NIVR and KNMI in the
Netherlands. We acknowledge the OMI International Science Team for producing OMI data. We
also acknowledge the ozonesonde providers and their funding agencies for making ozonesonde



measurements, and the Aura Validation Data Center (AVDC), WOUDC, SHADOZ, DISCOVER-
AQ, and SEACR$^4$S for archiving the ozonesonde data.



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





**Figures and Figure Captions**

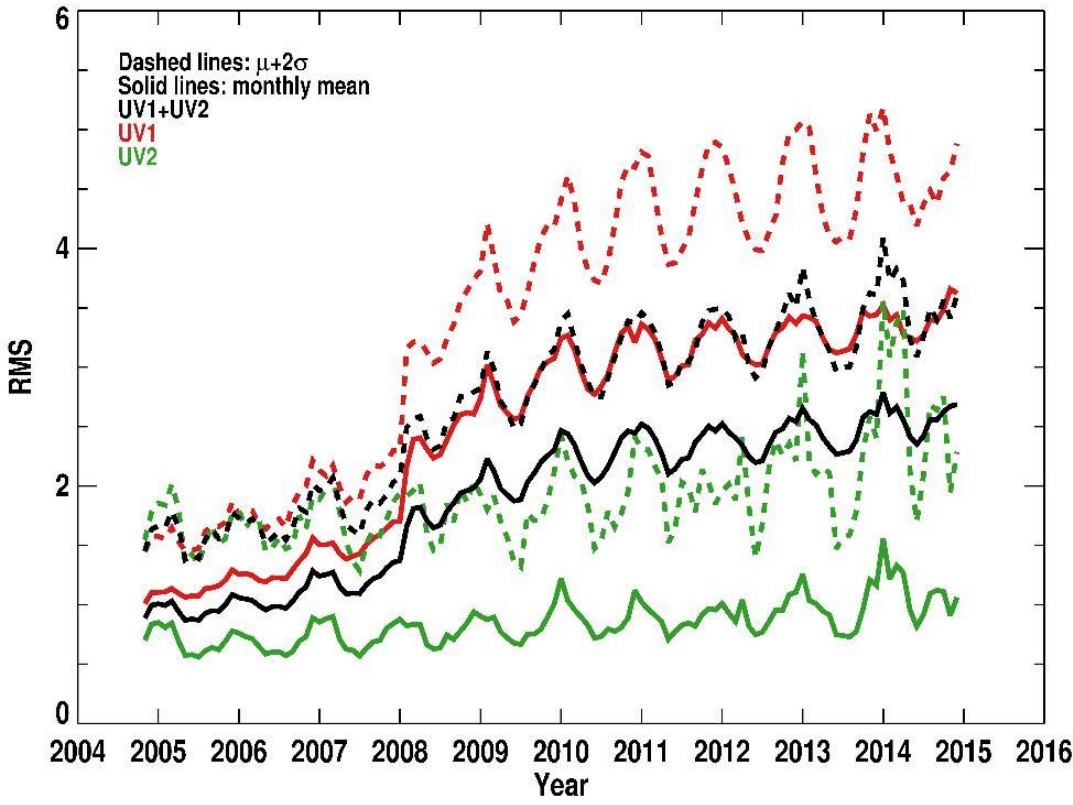


**Figure 1 Variation of monthly mean OMI RMS (defined as Root Mean Square of the ratio of radiance**
**residuals to assumed radiance errors). The dashed and solid lines represent respectively the monthly**
**mean RMS, and the sum of monthly mean plus its two standard deviations that is set as the RMS**
**threshold for data screening.**





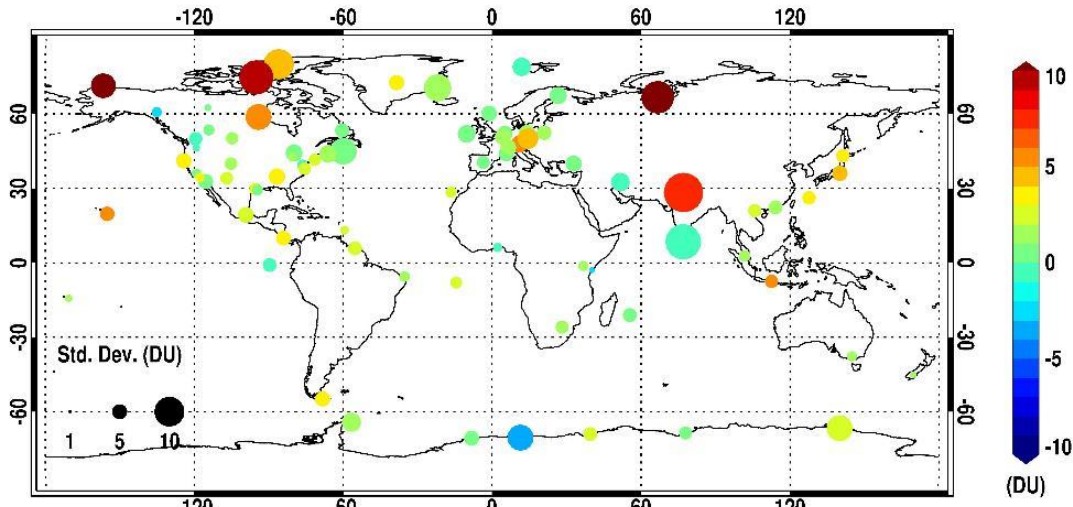


**Figure 2 The distribution of ozonesonde stations in this study. The color represents the mean biases**
**between OMI and ozonesonde tropospheric ozone columns (TOCs) at each station (if the number of**
**OMI and ozonesonde pairs is more than 10), and the dot size represents the standard deviation.**



**Figure 3 Mean relative biases in ozone (line with circles) and corresponding standard deviations (solid lines) between OMI retrieval/a priori and ozonesondes with and without applying OMI retrieval averaging kernels (i.e., with AKs, and W/O AKs in red and green for comparing retrievals and in blue and yellow for comparing a priori) for five different latitude bands. The left panels show the comparison using 10 years of OMI data (2004-2014), and the right panels show the comparison**





**between OMI retrieval and ozonesonde with OMI AKs for before and after the occurrence of serious**
**OMI row anomaly (RA), i.e., pre-RA (2004-2008) in black and post-RA (2009-2014) in gray,**
**respectively. The number (N) of OMI/ozonesonde coincidences used in the comparison is indicated**
**in the legends.**






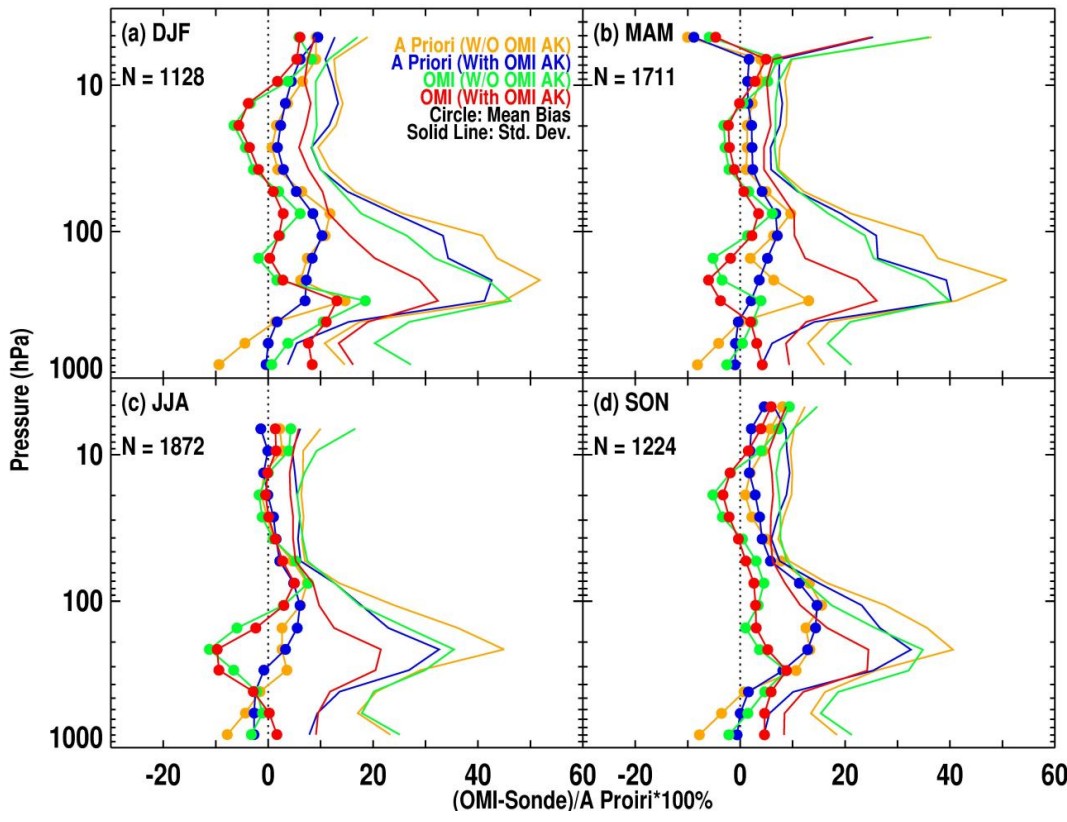


**Figure 4 Same as Figure 3c but for each individual season at 30° N-60° N.**






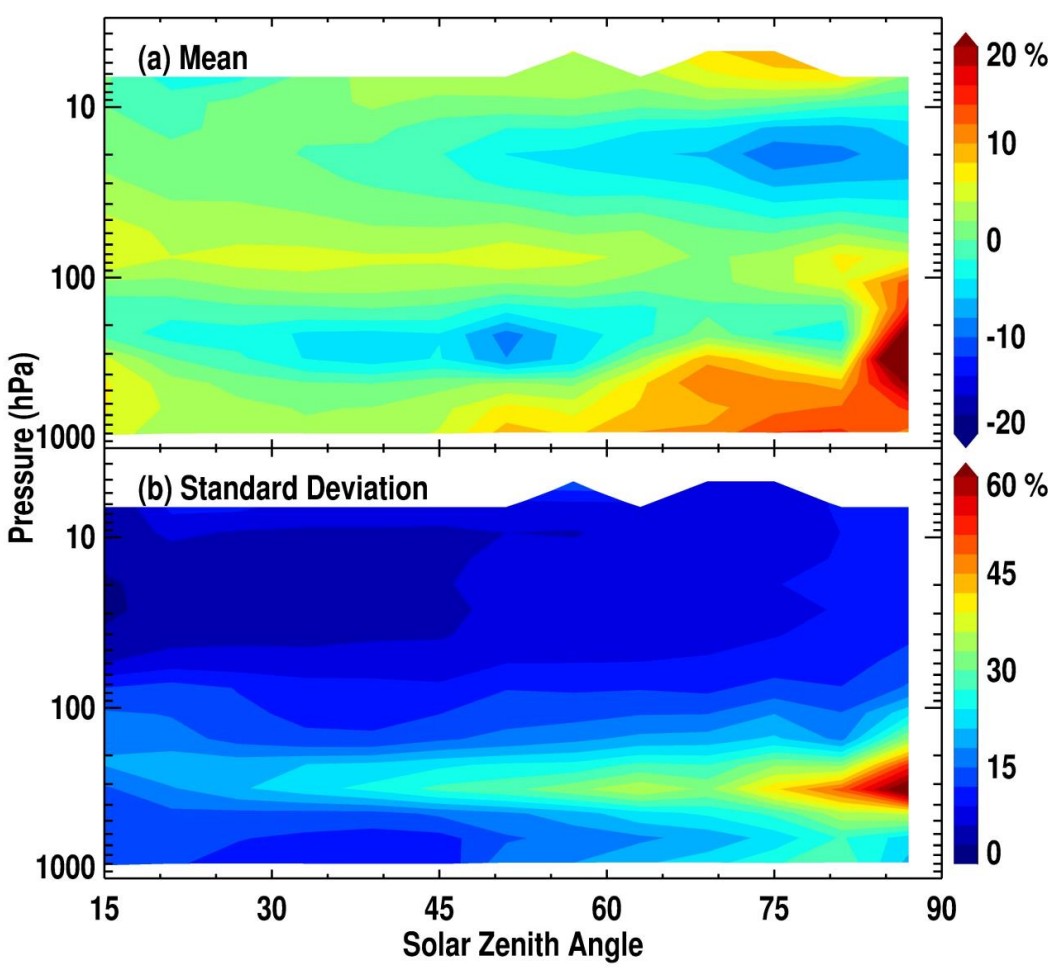


**Figure 5 Mean relative biases in ozone (a) and standard deviations (b) of the differences between OMI and ozonesonde convolved with OMI AKs as a function of Solar Zenith Angle using all OMI/ozonesonde coincidences during 2004-2014.**







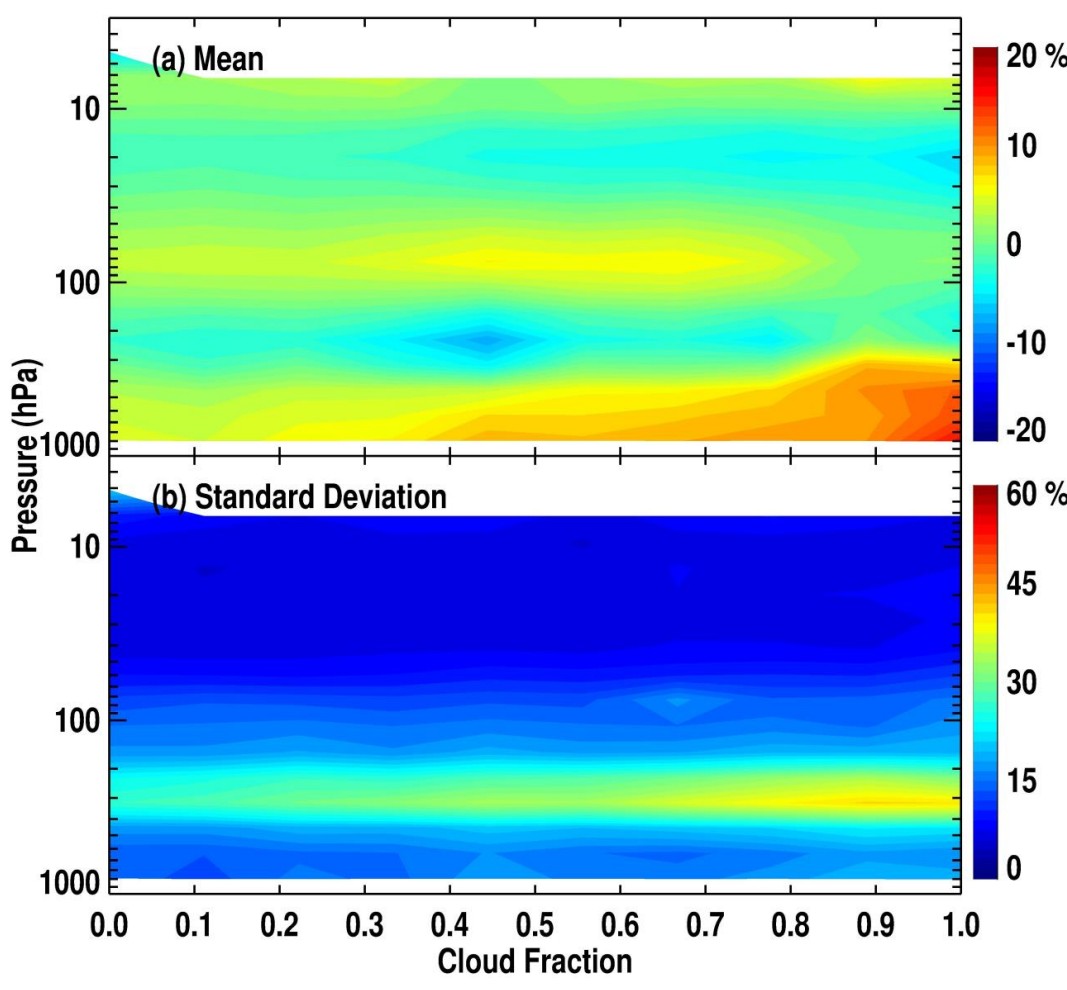


**Figure 6 Same as Figure 5 but as a function of cloud fraction.**






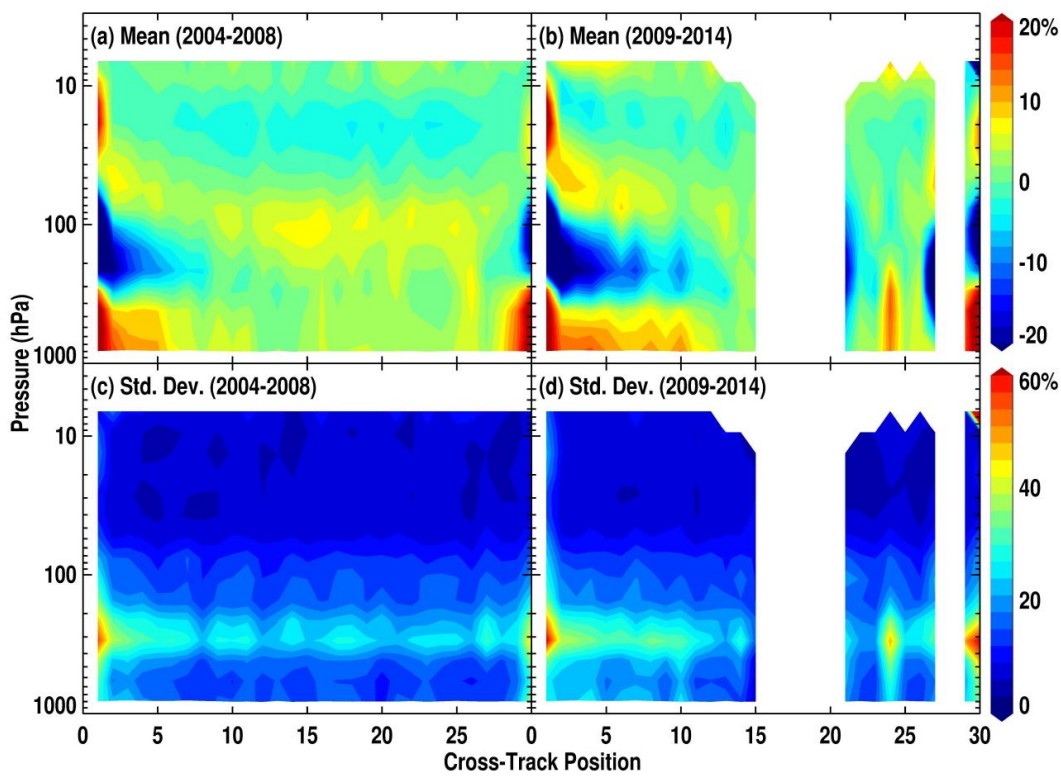


**Figure 7 Same as Figure 5 but as a function of cross-track position for (left) pre-RA (2004-2008) and (right) post-RA (2009-2014) periods, respectively.**






**Figure 8. Scattering plots of OMI Stratospheric Ozone Columns (SOCs) vs. ozonesonde SOCs without (black) and with (red) average kernels for five different latitude bands during 2004-2014 (left), the pre-row anomaly (RA) period (i.e., 2004-2008, middle) and the post-RA period (i.e., 2009-2014, right), respectively. Comparison statistics including mean biases and standard deviations in both DU and %, the linear regression and correlation coefficients in DU, and the number of coincidences are shown in the legends.**



963

**Figure 9. Similar to Figure 8, but for comparison of Tropospheric Ozone Columns (TOCs).**



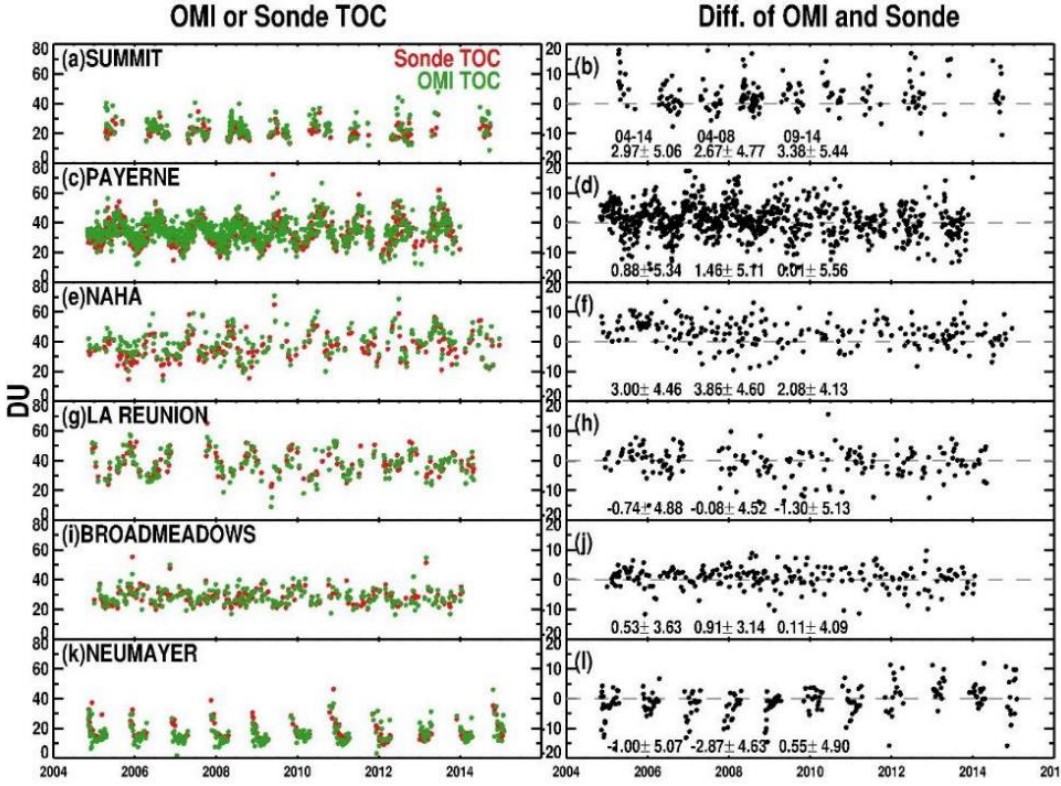

965

**Figure 10. (Left) Time series of OMI tropospheric ozone columns (TOCs) as green dots and
ozonesonde TOCs (with OMI AKs applied) in Summit (38.48° W, 72.57° N), Payene (6.57° E, 46.49°
N), Naha (127.69° E, 26.21° N), La Réunion (55.48° E, 21.06° S), Broadmeadows (144.95° E, 58.74°
S) and Neumayer (8.27° W, 70.68° S), and (Right) their corresponding differences, including the
mean biases and standard deviations in 2004-2014, pre-RA (2004-2008) and post-RA (2009-2014)
periods, respectively, in the legends.**






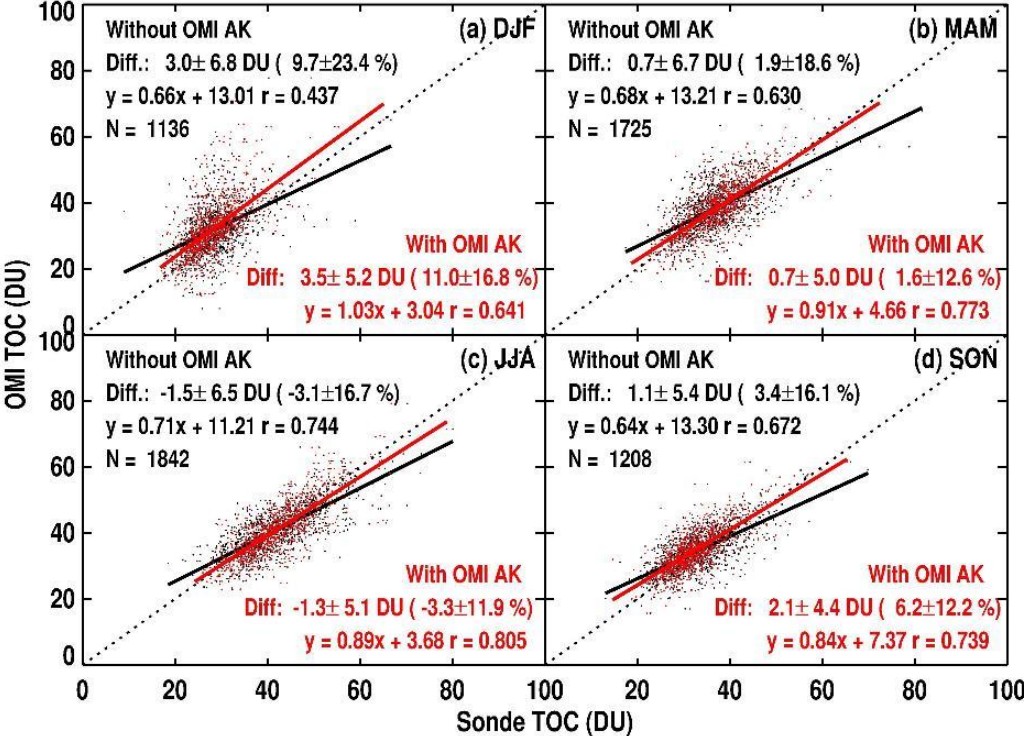


**Figure 11. Same as Figure 9 but for different seasons at northern middle latitude during the 2004-2014 period.**







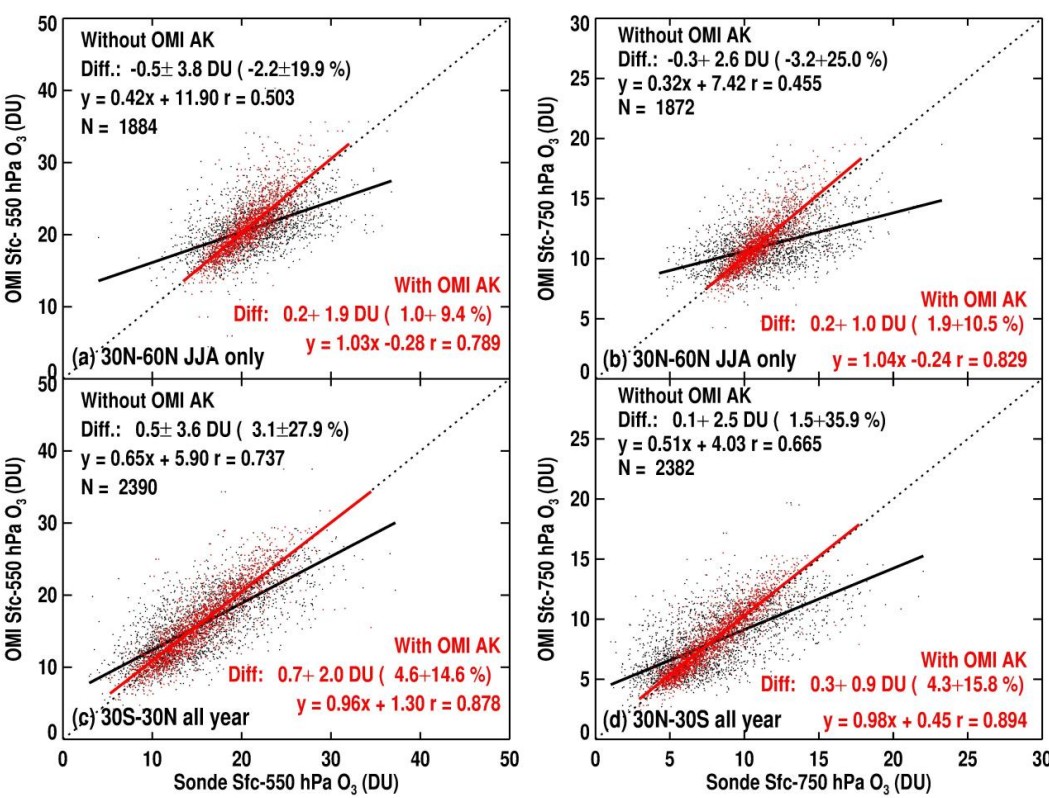


**Figure 12. Same as Figure 8 but for comparison of lower tropospheric ozone columns. (a) Surface~550 hPa ozone column and (b) Surface~750 hPa ozone column in 30° N-60° N during the summer, (c) and (d) same as (a) and (b) but for the tropics.**






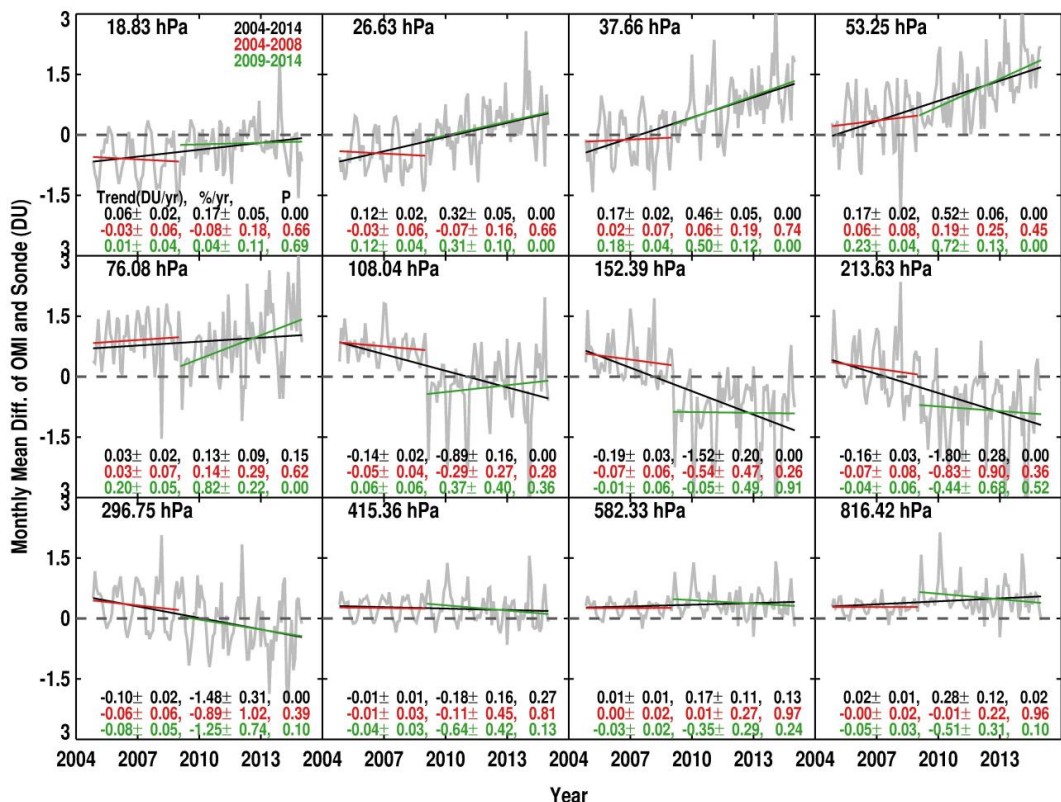


**Figure 13. Monthly mean variation of OMI and ozonesonde mean biases in 60° N-60° S at each OMI layer. The black, red and green lines represent the linear ozone bias trends in 2004-2014, pre-RA (2004-2008) and post-RA (2009-2014), respectively. The average altitude of each layer is marked on the left corner of each grid. The trends in DU/yr or % yr and P value for each time period are indicated in the legends.**

988





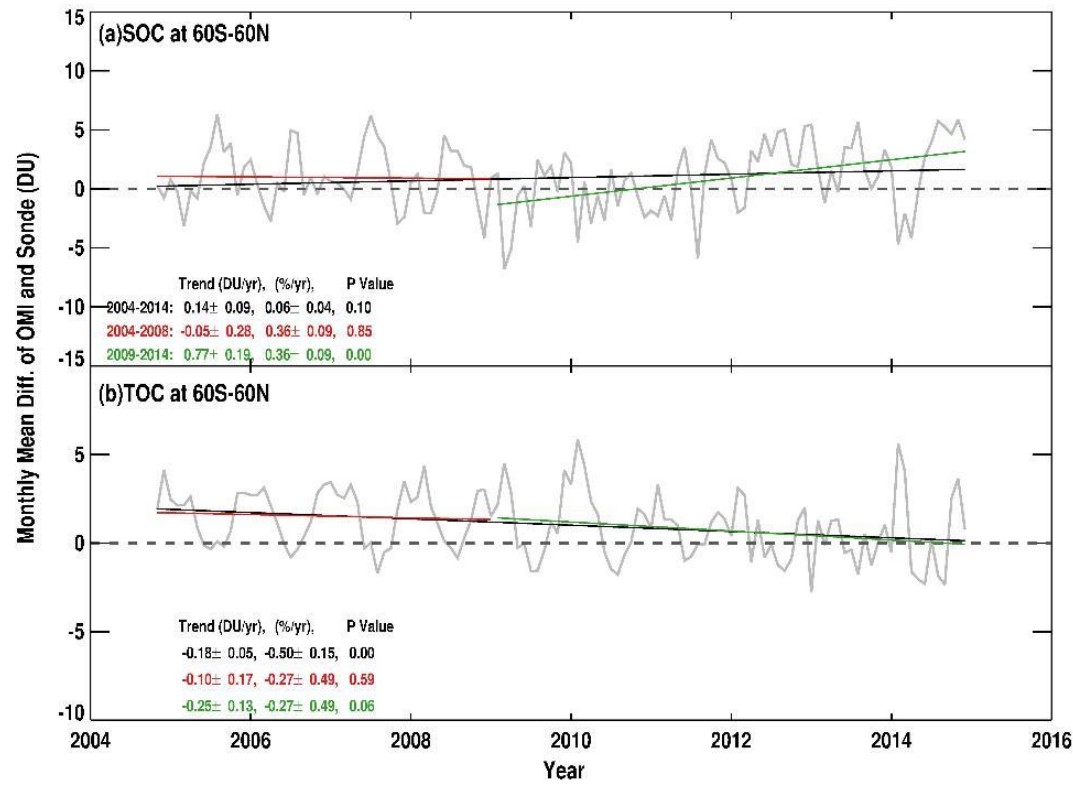

989

**Figure 14. Same as Figure 13 but for Stratospheric Ozone Columns (SOCs) and Tropospheric Ozone**
**Columns (TOCs).**