# Peer review of "Validation of 10-year SAO OMI Ozone Profile (PROFOZ) Product Using Ozonesonde Observations"

_Atmospheric Measurement Techniques, 2017_

## Referee Comment (RC1) · Anonymous Referee #1 · 23 Mar 2017

Overview:

The authors have produced a carefully determined retrieval of ozone profiles, SOCs, and TOCs for over 10 years from an OMI profile retrieval algorithm (SAO). They compared retrieved SOC, TOC, and profile measurements extensively with global ozonesondes. These ozonesonde comparisons included filtering the OMI measurements for nearly clear-sky scenes, SZA < 75 degrees, and cross-track positions least affected by OMI row anomaly. The authors show from differences between pre- and post-row anomaly periods that the current 10-year profile product (and derived columns) does not appear to be useful for evaluating decadal trends; however, the product at shorter timescales including daily from the analyses appears to be a useful

science product, particularly from the tropics to mid-latitudes. The paper appears good in current form and publishable with mostly just a few small comments that are listed below.

General comments:

* Lines 116-118: You might include in this reference list the paper by Yang et al. [2007] which used OMI and MLS to derive tropospheric ozone columns:

Yang, Q., D. M. Cunnold, H. –J. Wang, L. Froidevaux, H. Claude, J. Merrill, M. Newchurch, and S. J. Oltmans, Midlatitude tropospheric ozone columns derived from the Aura Ozone Monitoring Instrument and Microwave Limb Sounder measurements, J. Geophys. Res., 112, D20305, doi:10.1029/2007JD008528, 2007.

* Line 165: Were the NCEP tropopause pressures for getting TOCs and SOCs determined from a PV-theta definition, or a lapse rate definition, or something else?

Section 4.1.3.: Is your derivation of cloud optical centroid pressure and your effective scene pressure the same as Joiner et al. (2009, ACP)? Is there any major difference with your effective cloud fraction and their radiative cloud fraction for determining effective scene pressure? Is effective scene pressure determined the same?

Joiner, J., M. R. Schoeberl, A. P. Vasilkov, L. Oreopoulos, S. Platnick, N. J. Livesey, and P. F. Levelt (2009), Accurate satellite-derived estimates of the tropospheric ozone impact on the global radiation budget, Atmos. Chem. Phys., 9, 4447-4465, doi:10.5194/acp-9-4447-2009.

* Line 257: This appears to be a dead link.

* Line 274: There are more error sources than just the pump efficiency for the ozonesondes, but correct that pump efficiency errors are largest in higher altitudes.

* Where "SOC" is first mentioned (including the Abstract) it might be useful to state clearly that SOC is not the generally inferred total stratospheric ozone column but instead the ozone column from the tropopause up to balloon burst pressure. In your second submitted joint validation paper that uses MLS, "SOC" probably refers to total stratospheric ozone column?

* Line 343: Should be "...stratosphere (UTLS)..."

* Many of the figures, if intended as single column will have figure text that will be too small to read. The authors might specify to the journal that these figures should be printed double column, or perhaps instead increase some of the figure text.

---

## Referee Comment (RC2) · Anonymous Referee #2 · 26 Mar 2017

The manuscript deals with the validation of nadir (partial) ozone profiles from 10 years of OMI data using balloon soundings. The quality and limitations of the products are well described showing the results of various tests and for different conditions.

General comments:
To me it would have made the narrative order more logic to first read about the tests on SZA, cloud fraction and cross-track position dependence before presenting the results of the final dataset resulting from the selected criteria (i.e. move section 4.1.1 to 4.1.4).
Please mention version numbers throughout the text ('current', 'updated', etcetera are confusing).

Specific/technical comments:
References to Huang e.a., 2016: I am not sure if you can specify 2016 if it is not yet published in ACP discussions.

Page 5 line 105: remove second 'in' (.. retrieval errors in in the range..).
Page 9 line 215: there seems to be a space missing in '2009and'
Page 9 line 219, second criterion: rephrase, now it appear to be a contradiction that you select rows 4-27 but state that these have a worse quality and larger footprint as that is what this selection is avoiding.
Page 11 line 295 remove the second (double) comma after the first use of OMI
Page 12 line 311 add 'above' to "~30 km"
Page 12 lines 319-310. Why do you not use median/percentile values to deal with outliers?
Page 13 line 343: second troposphere should be stratosphere
Page 13 lines 347-348: '.. differences .. can be reduced ..". Unclear phrasing, do you mean that this could be an improvement in a future version or do you apply it here?
Page 14 line 389: underestimate → underestimates
Page 15 lines 400-401: I find it strange to read 'best agreement .. (except for the MBs) ..'. Given that the MBs are not closer to zero at all altitudes in the NH summer, please consider rephrasing 'best' (also in the conclusions section).
Page 15 lines 407-410: Please clarify that you are referring to SDs only here: the non NH summer seasons show improvements in MBs over the apriori only above the 3-4 lowermost layers (not 2-3). Also, the statement on the NH summer season improvements 'at all tropospheric layers except for the bottom one' is not valid for the MBs.
Page 17 line 462: I guess you mean striping instead of stripping ☺
Section 4.2.1 What pressure/altitude border do you use? How large is the contribution of the assumed profile above the burst altitude?
Page 19 line 513: missing space in '1-3%except'
Page 20 lines 546-547: why only at New Delhi, as Trivandrum also uses Indian sonde? What about the SD at Poona?
Page 20 line 549: 'no much' → 'little' or 'not much'
Page 21 lines 590-591: add references for this statement.
Page 24 line 664: 'the comparison is seasonally dependent'. Rephrase (the results may depend on season but the comparison shouldn't)

Figures 3 and 4: Label of the x-axis states 'A proiri' → A priori
Figure 3 (d, f, h): explain why sometimes the pre-/post RA number of collocations used (N) does not sum to the N for the full period (c, e, g).
Figure 6: the colour bars are distinct from similar figures (5 and 7) – esthetic, not a real problem ☺
Figures 8 to 12: these figures are quite fuzzy in the pdf, especially when zooming in to see the (colour of the) small dots – please check the figures' resolution for the final publication.
Figure 10: Consider that red versus green can be confusing for the colour blind. You might want to extend or shorten the vertical axis so that you get rid of the overlap between 0 and 80 DU.

Figure 11 caption states 'same as Figure 9' and Figure 12 caption states 'same as Figure 8'. Since the figures' setup refer to subsets and have a different orientation, I would rephrase this.

Figure 12 caption: is this for the full time series as for Figure 11?

Figure 12a states '30N-60 N JJA only' in the figure whereas it should be '30N-60N all year'.

Figure 13 caption: state use of AVK

Figure 14b: for consistency with 14a add the years to the numeric information on the trends in the figure?

---

## Author Comment (AC1) · 10 May 2017

Response to Referee #1

We thank referee's helpful and constructive comments and review. The referee's comments are listed in *italics*, and our responses in black with revised texts in **bold black**.

*Overview:*
*The authors have produced a carefully determined retrieval of ozone profiles, SOCs, and TOCs for over 10 years from an OMI profile retrieval algorithm (SAO). They compared retrieved SOC, TOC, and profile measurements extensively with global ozonesondes. These ozonesonde comparisons included filtering the OMI measurements for nearly clear-sky scenes, SZA < 75 degrees, and cross-track positions least affected by OMI row anomaly. The authors show from differences between pre and post-row anomaly periods that the current 10-year profile product (and derived columns) does not appear to be useful for evaluating decadal trends; however, the product at shorter timescales including daily from the analyses appears to be a useful C1 science product, particularly from the tropics to mid-latitudes. The paper appears good in current form and publishable with mostly just a few small comments that are listed below.*

*General comments:*
*\* Lines 116-118: You might include in this reference list the paper by Yang et al. [2007] which used OMI and MLS to derive tropospheric ozone columns: Yang, Q., D. M. Cunnold, H. –J. Wang, L. Froidevaux, H. Claude, J. Merrill, M. Newchurch, and S. J. Oltmans, Midlatitude tropospheric ozone columns derived from the Aura Ozone Monitoring Instrument and Microwave Limb Sounder measurements, J. Geophys. Res., 112, D20305, doi:10.1029/2007JD008528, 2007.*

Done.

*\* Line 165: Were the NCEP tropopause pressures for getting TOCs and SOCs determined from a PV-theta definition, or a lapse rate definition, or something else?*

The tropopause pressures are defined by the lapse rate. We have added "**(defined based on the lapse rate)**" after the tropopause.

*Section 4.1.3.: Is your derivation of cloud optical centroid pressure and your effective scene pressure the same as Joiner et al. (2009, ACP)? Is there any major difference with your effective cloud fraction and their radiative cloud fraction for determining effective scene pressure? Is effective scene pressure determined the same?*
*Joiner, J., M. R. Schoeberl, A. P. Vasilkov, L. Oreopoulos, S. Platnick, N. J. Livesey, and P. F. Levelt (2009), Accurate satellite-derived estimates of the tropospheric ozone impact on the global radiation budget, Atmos. Chem. Phys., 9, 4447-4465, doi:10.5194/acp-9-4447-2009.*

Yes, we use the optical centroid cloud pressure (OCCP) from the OMI Raman cloud product by J. Joiner (Vasilkov et al., 2008), i.e., same as in Joiner et al. (2009). We directly use cloud pressure values meeting all the recommended quality flags. For pixels without qualified OCCP values, they are filled in by spatial interpolation on an orbital basis. Remaining empty values after the interpolation are filled in by a climatology derived from 7 years of OMI OCCPs. We use pixel-independent approximation for partial cloudy conditions, i.e., Lambertian clouds with OCCP and clear-sky conditions with surface pressure, and thus we do not use effective scene pressure. We use the effective cloud fraction from the same cloud product as initial value. We re-derive the effective cloud pressure from radiances at a weakly-absorbing wavelength (~347 nm) similar to the Raman cloud product. Typically, the effective cloud-top pressure is used in the radiative transfer calculation. Radiative cloud fraction, ratio of cloud radiance to total radiance, is derived from the effective cloud fraction.

We have added "**from the OMI Raman cloud product (Vasilkov et al., 2008)**" after "Our OMI ozone algorithm assumes clouds as Lambertian surfaces with optical centroid cloud pressure"

*\* Line 257: This appears to be a dead link.*
The address is correct. But the line number of 258 is added between "discover-" and "aq" when copying the link or clicking it directly. We have corrected it the updated version.

*\* Line 274: There are more error sources than just the pump efficiency for the ozonesondes, but correct that pump efficiency errors are largest in higher altitudes.*

We have changed "uncertainties in pump efficiency" to "**uncertainties mainly from pump efficiency**"

*\*\* Where "SOC" is first mentioned (including the Abstract) it might be useful to state clearly that SOC is not the generally inferred total stratospheric ozone column but instead the ozone column from the tropopause up to balloon burst pressure. In your second submitted joint validation paper that uses MLS, "SOC" probably refers to total stratospheric ozone column?*
The SOC is first mentioned in the Abstract. We have revised it as:

"The MBs of the stratospheric ozone column (SOC**, the ozone column from the tropopause pressure to the ozonesonde burst pressure**) are within 2% with SDs of < 5% and the MBs of the tropospheric ozone column (TOC) are within 6% with SDs of 15%."

In addition, we have described the SOC as the ozone column from the tropopause pressure to the corresponding ozonesonde burst pressure in the Sect. 3. But to make it more clear, we have changed "The TOC is integrated from the surface to the tropopause and the SOC is integrated from the tropopause pressure to the ozonesonde burst pressure" to "The TOC is integrated from the surface to the tropopause. **And** the SOC is **not the total stratospheric ozone column, but the**

**ozone column** integrated from the tropopause pressure to the ozonesonde burst pressure."

*\* Line 343: Should be ". . .stratosphere (UTLS). . ."*
Done.

*\* Many of the figures, if intended as single column will have figure text that will be too small to read. The authors might specify to the journal that these figures should be printed double column, or perhaps instead increase some of the figure text.*

Thank for the suggestion. We will contact the journal about this during the production process.

References:

Vasilkov, A. P., J. Joiner, R. Spurr, P. K. Bhartia, P. F. Levelt, and G. Stephens: Evaluation of the OMI cloud pressures derived from rotational Raman scattering by comparisons with satellite data and radiative transfer simulations, J. Geophys. Res., 113, D15S19, doi:10.1029/2007JD008689, 2008.

---

## Author Comment (AC2) · 10 May 2017

Response to Referee #2

We thank referee's helpful and constructive comments and review. The referee's comments are listed in *italics*, and our responses in black with revised texts in **bold black**.

*The manuscript deals with the validation of nadir (partial) ozone profiles from 10 years of OMI data using balloon soundings. The quality and limitations of the products are well described showing the results of various tests and for different conditions.*

*General comments:*
*To me it would have made the narrative order more logic to first read about the tests on SZA, cloud fraction and cross-track position dependence before presenting the results of the final dataset resulting from the selected criteria (i.e. move section 4.1.1 to 4.1.4).*

We agree that it is more logical to first discuss tests on SZA, cloud fraction and cross-track position dependence before presenting the results of the general results. However, we have already mentioned conducting the tests on these parameters and the order in Sect. 2.1 and in Sect. 3, and the discussion of Sects. 4.1.2-4.1.4 depends on results and discussion in Sect. 4.1.1. Therefore, we keep our original order to present the overall results first.

To make it clear, we have changed "We will use all OMI pixels of each filtering parameter when evaluating retrieval quality as a function of that specific parameter." To "**The selection and justification of these criteria will be discussed in Sects. 4.1.2-4.1.4, in which w**e will use all OMI pixels of each filtering parameter when evaluating retrieval quality as a function of that specific parameter." in line 223 of Sect. 2.1 of the AMTD manuscript.

Also in section 3, we have changed "Although we filter OMI data based on cloud fraction, cross-track position, and SZA, we conduct the comparison as a function of these parameters using coincidences at all latitude bands to show how these parameters affect the retrieval quality" to "Although we filter OMI data based on cloud fraction, cross-track position, and SZA **in the final evaluation of our retrievals against ozonesonde observations as shown in Sect. 4.1.1**, we conduct the comparison as a function of these parameters using coincidences at all latitude bands to show how these parameters affect the retrieval quality **as shown in Sects. 4.1.2-4.1.4.**"

*Please mention version numbers throughout the text ('current', 'updated', etcetera are confusing).*
Unfortunately, SAO OMI retrieval algorithm does not have any official version number. The first version is described in Liu et al. (2010) and the current (with minor updates) version that is used in this paper is briefly described in Kim et al. (2013). The future or next version refers to the version that we are currently working on.
For clarifying, we have revised the description of our current algorithm in Set. 2.1 as follows:
"The **current** algorithm of our SAO OMI ozone product **that is used in this paper** was briefly described in Kim et al. (2013)**.**"

Specific/technical comments:
References to Huang et al., 2016: I am not sure if you can specify 2016 if it is not yet published in ACP discussions.

This paper had been submitted to Atmospheric Measurement Techniques and has been published at AMTD. We have updated the reference and citation as:

**"Huang, G., Liu, X., Chance, K., Yang, K., and Cai, Z.: Validation of 10-year SAO OMI Ozone Profile (PROFOZ) Product Using Aura MLS Measurements, Atmos. Meas. Tech. Discuss., 2017, 1-25, doi: 10.5194/amt-2017-92, 2017."**

*Page 5 line 105: remove second 'in' (.. retrieval errors in in the range.).*
Done.

*Page 9 line 215: there seems to be a space missing in '2009and'*
Done.

*Page 9 line 219, second criterion: rephrase, now it appear to be a contradiction that you select rows 4-27 but state that these have a worse quality and larger footprint as that is what this selection is avoiding.*
We have rephrased it as:
"… 2) cross track positions between 4 and 27**,** due to **the** relatively worse quality and much larger footprint size of **the off-nadir pixels beyond this range**; …"

*Page 11 line 295 remove the second (double) comma after the first use of OMI*
Done.

*Page 12 line 311 add 'above' to "~30 km"*
Done.

*Page 12 lines 319-310. Why do you not use median/percentile values to deal with outliers?*
It is a good idea to use the median/percentile values to deal with outliers. But it is also common to remove outliers beyond the range of means plus and minus 3 times of standard deviations. Switching to the use of median/percentile values should not affect the overall conclusion of this paper.

*Page 13 line 343: second troposphere should be stratosphere*
Done.

*Page 13 lines 347-348: '.. differences .. can be reduced ..". Unclear phrasing, do you mean that this could be an improvement in a future version or do you apply it here?*
Yes. We mean an improvement in a future version. We have rephrased it as:
"Consequently, the SDs of OMI/sonde differences in the UTLS at mid- and high-latitudes can be reduced through reducing the retrieval uncertainties **in a future version of the algorithm that uses the TB climatology**."

*Page 14 line 389: underestimate -> underestimates*
Done. We think you mean the "underestimate" at line 378.

*Page 15 lines 400-401: I find it strange to read 'best agreement .. (except for the MBs) ..'. Given that the MBs are not closer to zero at all altitudes in the NH summer, please consider rephrasing 'best' (also in the conclusions section).*

We refer "best agreement" to the smallest standard deviations. We have rephrased the sentence "The comparison results are clearly season-dependent with best agreement in the summer (except for the MBs) and the worst agreement in the winter" to "The comparison results are clearly season-dependent **with different altitude-dependent bias patterns, and** with **the smallest SDs** in the summer and the worst **SDs** in the winter"

In the conclusion, we have changed the original sentence "At northern mid-latitudes, the agreement is generally best (except for MBs) in the summer, with the best retrieval sensitivity and the smallest SDs as great as 20%, and the worst in the winter with the worst retrieval sensitivity and the largest SDs reaching 31%" to "At northern mid-latitudes, **there are generally the best retrieval sensitivity and the smallest SDs as great as 20% in the summer, and the worst retrieval sensitivity and the largest SDs reaching 31% in the winter**".

*Page 15 lines 407-410: Please clarify that you are referring to SDs only here: the non NH summer seasons show improvements in MBs over the a priori only above the 3-4 lowermost layers (not 2-3). Also, the statement on the NH summer season improvements 'at all tropospheric layers except for the bottom one' is not valid for the MBs.*
Yes, we are referring to SDs only. We have revised it as:
"Also, the retrieval in the summer shows the most improvements **in terms of reduction in SDs** over the a priori in the lower troposphere at all tropospheric layers except for the bottom layer, while the retrievals during other seasons show the improvement over a priori only above the lowermost two/three layers."

*Page 17 line 462: I guess you mean striping instead of stripping* □
Yes.

*Section 4.2.1 What pressure/altitude border do you use? How large is the contribution of the assumed profile above the burst altitude?*
As mentioned in Sect. 3, the SOC is integrated from the tropopause pressure to the ozonesonde burst pressure. Based on the ozonesonde data with coincident total ozone measurements, ozone column above burst altitude constitutes 6-33% of stratospheric ozone column (14% on average).

*Page 19 line 513: missing space in '1-3%except'*
Done.

*Page 20 lines 546-547: why only at New Delhi, as Trivandrum also uses Indian sonde? What about the SD at Poona?*
Although the bias at Trivandrum is small, there is a large SD at this station similar to that at New Delhi. Poona ozonesonde station is not included in this TOC comparison due to the small number ($< 10$) of ozonesonde-OMI pairs after applying our validation filters.

We have revised the sentence from "The large bias of $>6$ DU at New Delhi is likely associated with the large uncertainties of the Indian ozonesonde data" to "**In addition, there is a** large bias of $> 6$ DU at New Delhi. **The poor comparisons at these two stations are** likely associated with the large uncertainties of the Indian ozonesonde data."

*Page 20 line 549: 'no much' □ 'little' or 'not much'*

Done.

*Page 21 lines 590-591: add references for this statement.*
Sorry for the confusion. We meant evaluation in Sects. 4.1 and 4.2. Therefore, we have revised "Previous evaluation" to "**Comparisons in Sects. 4.1 and 4.2**"

*Page 24 line 664: 'the comparison is seasonally dependent'. Rephrase (the results may depend on season but the comparison shouldn't)*
We have revised it **to "**The comparison **results are** seasonally dependent.**"**

*Figures 3 and 4: Label of the x-axis states 'A proiri' -> A priori*
Done.

*Figure 3 (d, f, h): explain why sometimes the pre-/post RA number of collocations used (N) does not sum to the N for the full period (c, e, g).*
This is because we applied the outlier removal process to each comparison, which may cause slight differences between the sum of the number of pre/post-RA collocations and the number collocations for the full period.

Figure 6: the colour bars are distinct from similar figures (5 and 7) – esthetic, not a real problem.

Thanks for pointing this out. We made them consistent.

*Figures 8 to 12: these figures are quite fuzzy in the pdf, especially when zooming in to see the (colour of the) small dots – please check the figures' resolution for the final publication.*
Yes, we will upload our high-resolution figures for the final publication.

*Figure 10: Consider that red versus green can be confusing for the colour blind. You might want to extend or shorten the vertical axis so that you get rid of the overlap between 0 and 80 DU.*
We have changed the colors readable to everyone, as well as the vertical axis to get rid of the overlap between 0 and 80 accordingly. The new figure is shown as below:

[Figure]

*Figure 11 caption states 'same as Figure 9' and Figure 12 caption states 'same as Figure 8'. Since the figures' setup refer to subsets and have a different orientation, I would rephrase this.*

We have revised the captions of Figure 11 and 12 as:

**"Figure11. Similar to panels in Fig. 9 but for different seasons at northern middle latitude during the 2004-2014 period.**

**Figure 12. Similar to panels in Fig. 9 but for comparison of lower tropospheric ozone columns during the 2004-2014 period. (a) Surface~550 hPa ozone column and (b) Surface~750 hPa ozone column in 30° N-60° N during the summer only, (c) and (d) same as (a) and (b) but for the tropics (30°S-30°N) during all year."**

*Figure 12 caption: is this for the full time series as for Figure 11?*
Yes, Figure 12 is for the full time series as for Figure 11.
We have revised Figure 12 caption as shown above.

*Figure 12a states '30N-60N JJA only' in the figure whereas it should be '30N-60N all year'.*
The original text in the figure is correct. Summer at mid-latitudes and all the seasons in the tropics are the times/locations when and where there are good retrieval sensitivities to lower tropospheric ozone.

*Figure 13 caption: state use of AVK*

We applied AVKs in this figure. We have added it a sentence "**OMI retrieval averaging kernels are applied to ozonesonde data.**"

*Figure 14b: for consistency with 14a add the years to the numeric information on the trends in the figure?*
Done.

References:
Kim, P. S., Jacob, D. J., Liu, X., Warner, J. X., Yang, K., Chance, K., Thouret, V., and Nedelec, P.: Global ozone–CO correlations from OMI and AIRS: constraints on tropospheric ozone sources, Atmos. Chem. Phys., 13, 9321-9335, doi: 10.5194/acp-13-9321-2013, 2013.
Liu, X., Bhartia, P. K., Chance, K., Spurr, R. J. D., and Kurosu, T. P.: Ozone profile retrievals from the Ozone Monitoring Instrument, Atmos. Chem. Phys., 10, 2521-2537, doi: 10.5194/acp-10-2521-2010, 2010.

---

## Author Response (AR2)

We thank editor's comments. The editor's comments are listed in *italics*, and our responses in black with revised texts in **bold black**.

*Dear Authors,*
*The reviewers' comments have been well taken into account. However, the question of Reviewer #2 about the missing version number of the retrieval algorithm, brings me to the question whether the data product has a version number? And whether the data product is available to users? Could you specify that at the end of the paper?*

The version number of PROFOZ is 0.9.3. This data product is available to users at Aura Validation Data Center (AVDC).

We have added a data availability section at the end of this paper as:

**"OMI PROFOZ (version 0.9.3) used in this study is available to users at Aura Validation Data Center (AVDC) (https://avdc.gsfc.nasa.gov/index.php?site=1389025893&id=74)."**

In addition, we have updated our data link at first paragraph of the Introduction section as:

"The ozone profile product titled PROFOZ is publicly available at the Aura Validation Data Center (AVDC) (**https://avdc.gsfc.nasa.gov/index.php?site=1389025893&id=74**)."

*Textual corrections:*
*- Caption Figure 8: Scattering plot > Scatter plot*

Done.
*- Caption of Figures 11 and 12: "Fig. 9but " has a missing space*

Done.

[revised manuscript text omitted]